# Multi-parameter photon-by-photon hidden Markov modeling

Paul David Harris [1✉], Alessandra Narducci [2], Christian Gebhardt[2], Thorben Cordes[2], Shimon Weiss [3,4] & Eitan Lerner [1,5✉]

Single molecule Förster resonance energy transfer (smFRET) is a unique biophysical approach for studying conformational dynamics in biomacromolecules. Photon-by-photon hidden Markov modeling (H$^2$MM) is an analysis tool that can quantify FRET dynamics of single biomolecules, even if they occur on the sub-millisecond timescale. However, dye photophysical transitions intertwined with FRET dynamics may cause artifacts. Here, we introduce multi-parameter H$^2$MM (mpH$^2$MM), which assists in identifying FRET dynamics based on simultaneous observation of multiple experimentally-derived parameters. We show the importance of using mpH$^2$MM to decouple FRET dynamics caused by conformational changes from photophysical transitions in confocal-based smFRET measurements of a DNA hairpin, the maltose binding protein, MalE, and the type-III secretion system effector, YopO, from *Yersinia* species, all exhibiting conformational dynamics ranging from the sub-second to microsecond timescales. Overall, we show that using mpH$^2$MM facilitates the identification and quantification of biomolecular sub-populations and their origin.

[1] Department of Biological Chemistry, The Alexander Silberman Institute of Life Sciences, Faculty of Mathematics & Science, The Edmond J. Safra Campus, The Hebrew University of Jerusalem, Jerusalem 9190401, Israel. [2] Physical and Synthetic Biology. Faculty of Biology, Ludwig-Maximilians-Universität München, Großhadernerstr. 2-4, 82152 Planegg-Martinsried, Germany. [3] Department of Chemistry and Biochemistry, and Department of Physiology, University of California, Los Angeles, CA, USA. [4] CaliforniaNanoSystems Institute, University of California, Los Angeles, CA, USA. [5] The Center for Nanoscience and Nanotechnology, The Hebrew University of Jerusalem, Jerusalem 9190401, Israel. ✉email: paul.harris@mail.huji.ac.il; eitan.lerner@mail.huji.ac.il

The role of structural dynamics in biomolecular function has come to the forefront of biophysical research[1,2]. Biomolecules in solution exhibit structural dynamics at a hierarchy of timescales and modes, from bond rotations to movements of entire globular domains, occurring at times from picoseconds to seconds and longer[3]. In many cases, the stages in the biomolecular function are promoted by different sub-populations of closely-related structures, or conformations. Examples include coupling of catalytic activity to domain dynamics in some enzymes[4,5], the dynamics of the DNA bubble in transcription initiation to support transcription start site selection[6,7], DNA mismatch repair[8], protein translocation[9], chaperone action[10], the allosteric regulation of the AAA+ disaggregase[11], active membrane transport[12–17], and many other important biochemical processes, in which structural dynamics is coupled to or influences biological function[1,2]. Thus, methods capable of identifying and characterizing distinctly time-separated structural sub-populations of biomolecules are of great interest in biomolecular sciences and structural biology.

NMR- and EPR-based methods[18–21] as well as single molecule methods[22–26] have come to the forefront in the field of dynamic structural biology, each with their own advantages and limitations. Single molecule methods allow probing one biomolecule at a time while tracking multiple experimental parameters simultaneously. This approach provides access to conformational heterogeneity, real-time kinetics, and identification of rare conformational states otherwise masked due to ensemble averaging.

One of the most popular single molecule approaches relies on the phenomenon of Förster resonance energy transfer (FRET), single molecule FRET (smFRET)[27], where the biomolecule of interest is site-specifically labeled at two strategic residues with two fluorescent dyes, which can exhibit transfer of excitation energy from the donor dye to the acceptor dye with a probability (or efficiency; $E$), which is inversely proportional to the sixth power of the distance between the dyes, according to the Förster relation[28–30]. The FRET efficiency can be determined either ratiometrically, through the donor and acceptor fluorescence intensities, or through the use of fluorescence lifetime-based methods. Ratiometric methods yield an initial raw efficiency, $E_{raw}$ (see Supplementary Eq. 1), to which correction factors must be applied, such as leakage of donor photons into the acceptor channel, direct excitation of the acceptor by the donor light source, differences in donor and acceptor fluorescence quantum yields and detection efficiencies (better known as the $\gamma$ -factor), in order to yield accurate $E$[31–33]. Lifetime-based approaches do not require such corrections, but rely on pulsed laser sources and time-correlated single photon counting modules[34]. SmFRET has proven to be a powerful tool to disentangle conformational sub-populations of biomacromolecules undergoing dynamic transitions over a range of timescales[3]. Nevertheless, smFRET remains limited by the time resolution and observation time of the apparatus[3]. A popular approach is the observation of individual freely-diffusing molecules through the excitation volume of a confocal microscope[1,2]. Here the observation time of a single molecule is on the order of a few milliseconds, with possible time resolution of dynamics as rapid as nanoseconds using advanced analyses of photon statistics within single molecule photon bursts (Fig. 1a, b). Some of the latter methods include photon distribution analysis, or probability distribution analysis (PDA)[35–40], burst variance analysis (BVA)[41], FRET two-kernel density estimator (FRET-2CDE)[42], analysis of two-dimensional histograms of donor fluorescence lifetimes, and ratiometric FRET efficiencies of bursts, also known as FRET lines[34,43,44], fluorescence correlation spectroscopy (FCS)[45,46] coupled to FRET[47–49], maximum likelihood approaches[50–54], such as hidden Markov modeling[4,7,55,56] (HMM), and photon recoloring[57,58]. These have been summarized in recent reviews of the field[1,2].

Photon-by-photon hidden Markov modeling (H²MM)[56] is a maximum likelihood method[57,59] that adopts the HMM machinery, while working directly with the photon data without binning into fluorescence intensity time traces, other than the clock time of the acquisition card, e.g. 50 ns for nsALEX, 12.5 ns

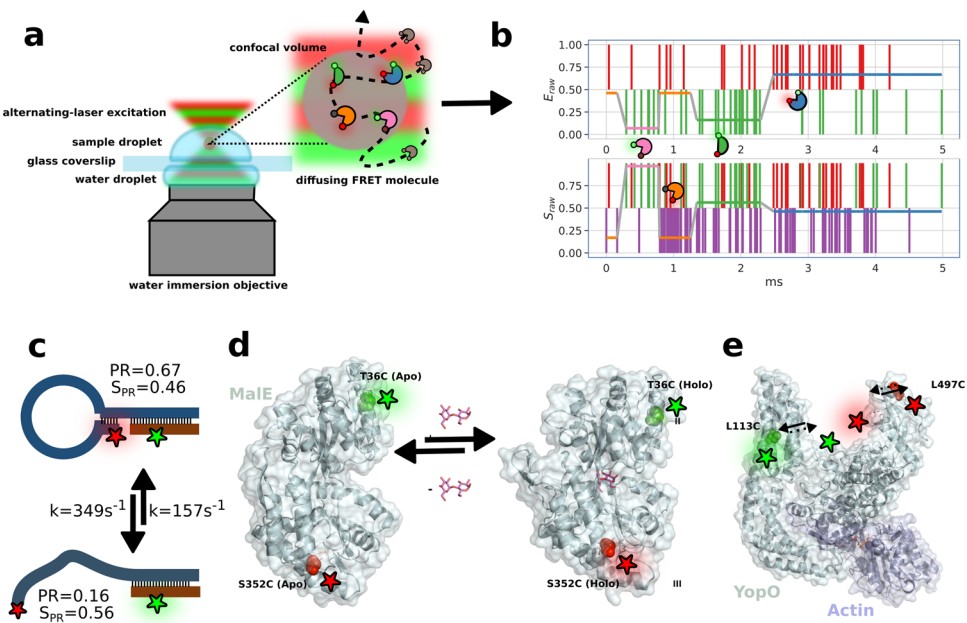

**Fig. 1 Cartoon representations of data acquisition, and biological systems examined in this work. a** Confocal microscope setup with inset illustrating the diffusive trajectory of a single molecule in and out of the confocal volume, undergoing conformational and photophysical changes, producing, **b** a photon time trace; photons represented by vertical bars, and the most likely state path according to the *Viterbi* algorithm overlaid as horizontal colored line. **c**–**e** Biological systems studied: **c** DNA hairpin, **d** maltose binding protein MalE with structure-guided conformational changes, and **e** type III secretion effector YopO. See Supplementary Fig. 1 for a version of this figure with transition rates, $E_{raw}$ and $S_{raw}$ values included for the biological systems.

for $\mu$sALEX. H²MM can extract the number of states involved in the underlying FRET dynamics, their mean $E_{raw}$ values and transition rate constants. Nevertheless, while advanced smFRET setups often detect multiple fluorescence parameters beyond the intensities, such as in alternating laser excitation (ALEX)[60,61] or in multi-color smFRET-based measurements[62–69], H²MM in its current iteration only uses the raw FRET efficiency of a single donor-acceptor pair of dyes.

Here, we introduce multi-parameter H²MM (mpH²MM), which enables incorporation of multiple parameters in the analysis, through additional photon streams. We demonstrate this concept with two types of ALEX experiments: microsecond ALEX ($\mu$sALEX) and nanosecond ALEX (nsALEX; known also as pulsed interleaved excitation, PIE)[60,61]. We applied this approach to different biomacromolecular complexes with dynamics ranging from the sub-second to microsecond timescales: (i) a DNA hairpin loop[70], (ii) the maltose binding protein MalE from E. coli, and (iii) YopO, a type-III-secretion system effector from pathogenic *Yersinia* species[71] (Fig. 1c–e, Supplementary Fig. 1). Our results and analysis demonstrate that mpH²MM is able to quantitatively report sub-populations based on both the ALEX-relevant mean parameters, $E_{raw}$ and the stoichiometry, $S_{raw}$ (see Supplementary Eq. 2), as well as their transition rate constants, demonstrating FRET-relevant conformational transitions, as well as FRET-irrelevant photophysical transitions. We also present the H2MM_C python package[72], with a backend written in C, for data processing, which is approximately two orders of magnitude faster than the previous implementation of H²MM in matlab[56].

Importantly, throughout this work, we make the clear distinction between sub-populations and states, where the latter is referred to the state models used to describe the dynamically interconverting sub-populations resolved from the data. This distinction is important, since thermodynamic states are single potential wells, and it is possible that the identified sub-populations are actually a group of states that interconvert much faster than the time resolution of the measurements. It should also be noted that we use the term parameter in multi-parameter H²MM to refer to parameters derived from ratios of sums of photons in different photon steams (e.g., E and S). These are distinct from state model parameters (e.g., rate constants, mean E).

## Results

### Verification of mpH²MM against simulated data

Analysis with single parameter H²MM (spH²MM) and mpH²MM can be performed using any given state model. Therefore, we must select the most likely state model among several, differing in their number of states and number of transition rate constants. Discriminating over- and under-fitted state models from the most likely model has proven difficult in the past[7,73]. Previously, we proposed the modified Bayes information criterion (BIC'), which does not provide an extremum-based decision on the most likely state model[7]. In the current work, we implement the integrated complete likelihood (ICL)[74,75], which gets a minimum value for the most likely state model, as the primary criterion for state model selection.

Using simulated smFRET data, where the ground truth of the number and properties of the states is known, we find that the ICL is more reliable than the BIC' at selecting the most reliable state model (see Supplementary Fig. 2, and Jupyter notebooks in supplementary dataset[72]). Yet, there are instances in the simulated data, and in real data sets, we describe later, where the selection of the most likely state model based on ICL is of a model with too few states, relative to our prior knowledge of the system. Therefore, we always consider the ICL first, then BIC', and take into account the prior knowledge of the system when selecting the most likely state model (see Supplementary Note 2 for expanded discussion, Supplementary Fig. 2).

To verify the validity of the multi-parameter approach, we perform a series of simple simulations (see supplementary Jupyter notebook mpH2MMsimulations[76]). We compare results of spH²MM and mpH²MM analyses of simulated data where the acceptor excitation photon stream was either included or excluded. Using this data, we find that selecting the most likely state model based on the ICL parameter reliably identifies the correct ground truth state model, and this model accurately reproduces the transition rate constants, $E_{raw}$ and $S_{raw}$ values used in the simulation (Supplementary Figs. 3 and 4, $E_{raw}$, Supplementary Table 1, $S_{raw}$ values defined in Supplementary Eqs. 6 and 7, respectively). In contrast, spH²MM is less reliable, and depending on the circumstances, it is unable to distinguish states with similar $E_{raw}$ values, which are easily distinguished in mpH²MM by their $S_{raw}$ values. Further, without the information about $S_{raw}$, interpretation of the models is more difficult, even if the correct number of states and their accurate $E_{raw}$ values are recovered in spH²MM.

### DNA hairpin exhibiting millisecond dynamics

As a first biological test system for mpH²MM, we used a DNA hairpin system introduced by Tsukanov et al. with a loop containing 31 adenines and a six base-pair stem[70]. The opening and closing rate constants of the hairpin vary as a function of the GC content of the stem as well as the sodium chloride (NaCl) concentration[70]. When appropriately labeled with a FRET donor and acceptor pair of dyes (ATTO 550 and ATTO 647N, respectively), the open and closed hairpin sub-populations exhibit distinct low and high mean $E_{raw}$ values, respectively. The hairpin containing two GCs out of the six stem bases, which we term HP3, exhibited opening and closing rates of a few milliseconds, depending on the NaCl concentration in the buffer. Such a DNA construct with well-characterized and tunable transition rates serves as an ideal model system to test and characterize the performance of mpH²MM.

We first perform nsALEX measurements[61] with this construct at a concentration of 300 mM NaCl, where a mix of both open and closed states are expected to interchange dynamically[70]. As a qualitative test for FRET dynamics occurring within bursts, we use burst variance analysis (BVA)[41], which compares the expected variance in $E_{raw}$ based on shot noise (the static FRET semi-circle) against the actual variance in $E_{raw}$. BVA of the HP3 data shows clear deviation from the static FRET semi-circle, suggesting that individual HP3 molecules are undergoing FRET dynamics as they traverse the confocal volume, which we term within-burst FRET dynamics (Fig. 2a). E-$\tau_D$ plots[44] also indicate within-burst dynamics (see Supplementary Fig. 5). However, without the prior knowledge of the DNA hairpin behavior as a two-state FRET system, and without knowing how many more sub-populations unrelated to FRET may exist, it is not necessarily clear how many distinct sub-populations are involved in within-burst dynamics. In visual examination of the 2D E-S plot, three sub-populations are apparent: (i) an open hairpin sub-population with mean $E_{raw}$ of 0.2, (ii) a closed hairpin sub-population, with a $E_{raw}$ of 0.65, both open and closed sub-populations have mean $S_{raw}$ of 0.5, and (iii) a third sub-population with a mean $E_{raw}$ of 0, and mean $S_{raw}$ of 1, where the acceptor is either in a dark state, or missing altogether (Fig. 2b). The 2D E-S plot also exhibits bursts with intermediate $E_{raw}$ values, bridging between the open and closed hairpin sub-populations. As these bursts are particularly dynamic in the BVA analysis, these are bursts where the hairpin is undergoing opening and closing transitions while crossing the confocal volume.

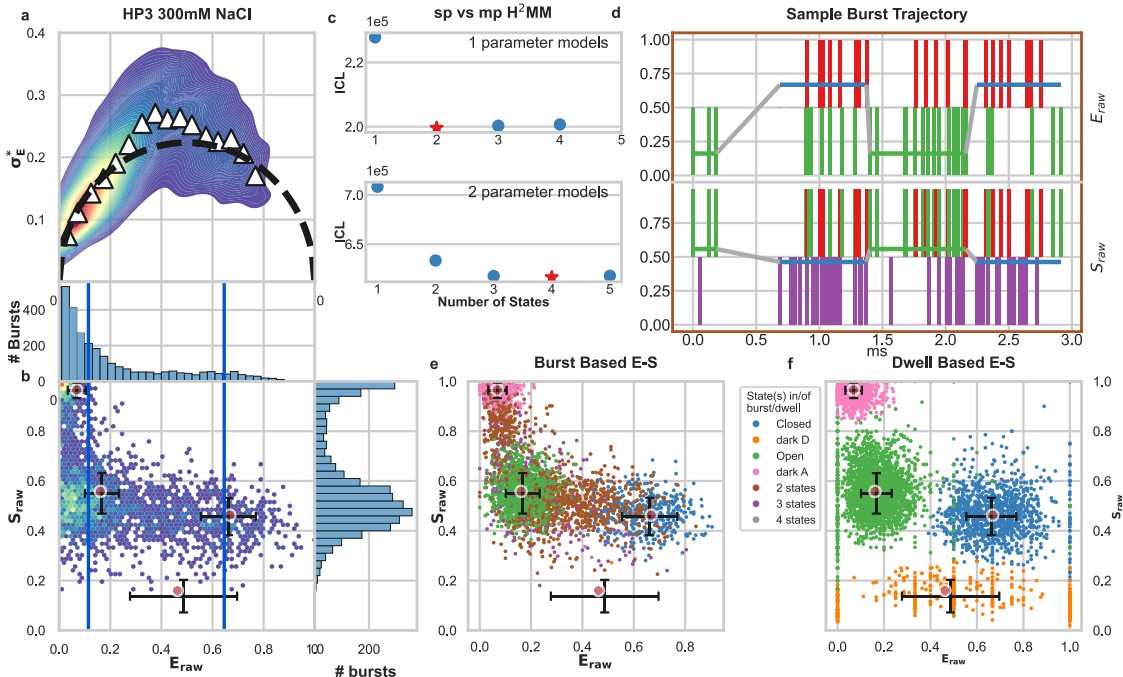

**Fig. 2 mpH$^2$MM results for DNA hairpin at 300 mM NaCl. a** Burst variance analysis (BVA), the $E_{raw}$ standard deviation of $E_{raw}$ values of bursts is displayed versus their $E_{raw}$ values. Bursts with $E_{raw}$ standard deviations higher than expected solely from shot noise (semi-circle), indicate dynamic heterogeneity, such as within-burst FRET dynamics. Triangles indicate the average of standard deviation values per $E_{raw}$ bin. **b** 2D histogram of $E_{raw}$ and $S_{raw}$ (E-S plots, colloquially) of bursts. The $E_{raw}$ and $S_{raw}$ values of sub-populations derived from mpH$^2$MM are marked by red circles, and the $E_{raw}$ and $S_{raw}$ standard deviation of these values, derived from the *Viterbi* dwell time analysis, are marked by black crosses. **c** Comparison of values of the integrated complete likelihood (ICL) of spH$^2$MM (top panel) and mpH$^2$MM (bottom panel) of optimized models with different state models. The most likely state model is marked as a red star. **d** A sample burst trajectory, with photons represented as colored vertical bars, with donor excitation photons colored green and red for donor and acceptor, respectively. Acceptor excitation photons are colored purple. $E_{raw}$ (top panel) and $S_{raw}$ (bottom panel) of sub-populations determined from dwells using the *Viterbi* algorithm, are overlaid on the photon bars, and colored to indicate the state of the dwell. The border color also represents the type of burst. **e, f** E-S scatter plots of data processed by the *Viterbi* algorithm. Consecutive photons with the same state are considered as a single dwell, $E_{raw}$ and $S_{raw}$ values are then calculated as in Supplementary Eqs. 8 and 9, respectively. MpH$^2$MM-derived sub-populations and *Viterbi*-derived $E_{raw}$ and $S_{raw}$ standard deviations (SD) are overlaid as red circles and black crosses, respectively. **e, f** E-S scatter plot of bursts **e** or dwells within bursts **f**, color coded by which states are present in the bursts (**e**) or according to the state of the dwell (dwell based $E_{raw}$ and $S_{raw}$ defined in Supplementary Eqs. 8 and 9, respectively) **f**, according to *Viterbi* algorithm. Color coding is the same throughout **d**, **e**, and **f**. Error bars (s.d.) in **b**, **e**, and **f** are the same, with $n = 1232$, 466, 3033, and 1493 dwells for the closed, dark donor, open, and dark acceptor states, respectively. See E-$\tau_D$ analysis in Supplementary Fig. 5, and more examples of bursts classified by the *Viterbi* algorithm in Supplementary Fig. 6.

Analyses of this data with spH$^2$MM and mpH$^2$MM show different patterns in the ICL values of the state models. The ICL is minimized for spH$^2$MM models for a two-state model, while it is minimized for a four-state model when using mpH$^2$MM. Visual inspection of the one-dimensional projection of burst data onto the $E_{raw}$ parameter immediately suggests an explanation for this discrepancy, as it appears as only two sub-populations. The donor-only or dark acceptor state, and the open hairpin state exhibit similar low $E_{raw}$ values and are difficult to distinguish as sub-populations based solely on $E_{raw}$. This projection reflects the data accessible to spH$^2$MM, the donor excitation streams, and thus the open hairpin and dark acceptor states are expected to have nearly identical FRET signatures with regard to the streams accessible to spH$^2$MM, thus leading to the false inference of only two states. The open hairpin FRET sub-population and the dark acceptor states are, however, quite distinct with regard to the acceptor excitation stream, which is accessible to mpH$^2$MM.

In the ICL-based selected four-state model retrieved by mpH$^2$MM, two out of the four states match nicely with the states in the ICL-based selected model from spH$^2$MM model, having similar $E_{raw}$ values. Their $S_{raw}$ values are ~ 0.5 (Fig. 2b red circles), as expected for molecules undergoing FRET. The third and fourth states in the model can be matched to dark acceptor and dark donor sub-populations, respectively. The third state has a $E_{raw}$ value ~ 0

and a $S_{raw}$ value ~ 1 (Fig. 2b, top left red circle). This state has a clear sub-population of bursts associated with it in the 2D E-S plot. The fourth state has an intermediate $E_{raw}$ value, and a very low $S_{raw}$ value of ~ 0.17 (Fig. 2b, bottom red circle, Supplementary Table 2). There is no obvious sub-population visually observed in the E-S plots to which this would correspond, but the $E_{raw}$ and $S_{raw}$ values are consistent with this being a dark donor state. More importantly, comparing the parameters of the state models retrieved by spH$^2$MM and mpH$^2$MM, we find that the transition rate constants derived using mpH$^2$MM are closer to those found by Tsukanov et al.[70] than those extracted using spH$^2$MM (Supplementary Table 3, and supplementary .csv files of all state models found by H$^2$MM analysis[72]). The transition rate constants provide a clue as to why the fourth state does not show up in the E-S plots as a distinct sub-population, as the transition rates predict rare transitions to it, and rapid transitions away from it. Thus, populating the fourth state occurs only briefly and rarely in bursts undergoing rapid dynamics, such that it does not appear as a clear sub-population in the E-S plots (Supplementary Table 3, and supplementary .csv file[72]).

The *Viterbi* algorithm finds the most likely state path through each burst, given a state model and its parameter values (Fig. 2d and Supplementary Fig. 6a–e). We use this to classify bursts by which states are present within each burst (Fig. 2d, Supplementary Fig. 6f), and separate photons into dwells, for which $E_{raw}$ and

$S_{raw}$ can be defined (Fig. 2d, Supplementary Eqs. 8 and 9 in Supplementary Note 1.3). Additional analysis of dwells and their durations is provided in Supplementary Fig. 7. Visual examination of the burst-based E-S plot (Fig. 2e) shows that the *Viterbi* algorithm reasonably classifies most bursts that have $E_{raw}$ and $S_{raw}$ values close to the predicted value of a given sub-population as only having that state present, as well as bursts with intermediate $E_{raw}$ and $S_{raw}$ that are predicted to include dwells of multiple states. Notably, there are only a few bursts classified as having dwells solely in the dark donor state (Fig. 2e), keeping with what is predicted by the transition rates, and indeed, few dwells are even found in this state (Fig. 2f, Supplementary Fig. 6g). The scarcity of the donor dark state in the *Viterbi* analysis serves to both confirm this observation and prove the sensitivity of mpH²MM at the same time. In summary, using spH²MM, we do not properly decouple the FRET-relevant information from the FRET-irrelevant dye transitions to fluorophore dark states for the DNA hairpin data, which influences the accuracy of the retrieved values for the $E_{raw}$ and rate constant parameters. On the other hand, using mpH²MM assists in the proper decoupling of the FRET-relevant information from the FRET-irrelevant ones and in gaining accurate parameter values. See Supplementary Figs. 8–19 for additional hairpin data acquired at different concentrations of NaCl. Now that we have verified mpH²MM with a well-defined biomolecular system of the DNA hairpin, HP3, we move to explore its usefulness in other biomacromolecular systems.

**Quantifying the dynamics of a substrate-binding protein.** In the previous example, we examined a system that exhibits intrinsic conformational dynamics, hence dynamics that is not induced by binding of a ligand. Now, we test mpH²MM on a system with conformational dynamics that is induced by substrate binding. For this, we select the periplasmic maltose binding protein, MalE from E. coli[77], which is the extracellular component of the maltose ABC importer MalFGK₂-E[77]. MalE is a bilobed protein with a structural core built from a periplasmic binding protein (PBP)-like II domain. Two rigid domains, $D_1$ and $D_2$, are separated by a two-segment $\beta$-strand hinge and are complemented by a C-terminal embellishment that facilitates structural dynamics between open and closed states[17]. This allows for MalE to close upon substrate binding, similar to a venus fly-trap. For our nsALEX smFRET measurements we produced a MalE double-cysteine variant with labels at the outer sides of the two lobes, specifically residues T36C and S352C. As shown previously, this enables tracking of the opening and closing dynamics in single MalE molecules[17,78]. We test three concentrations of the substrate maltose: none (apo), 1 $\mu$M (close to the $K_D$ value[77]) and 1 mM (holo). FRET histograms, using a dual channel burst search (DCBS)[36] filter exhibit three sub-populations: (i) a minor, low $E_{raw}$ sub-population at $E_{raw}$ of 0.1, (ii) a major sub-population with an intermediate $E_{raw}$ of 0.5, and (iii) a major sub-population with a high $E_{raw}$ of 0.7. We use DCBS because the donor- and acceptor-only sub-populations are very strong, and otherwise overwhelm the nsALEX data. Since we apply DCBS, bursts of the high $S_{raw}$ and low $E_{raw}$ values cannot represent molecules with permanently dark acceptor, but could be the result of either a real conformation, or of frequent acceptor blinking. With increasing maltose concentration, the fraction of the ~ 0.5 $E_{raw}$ sub-population decreases, while the fraction of the ~ 0.7 $E_{raw}$ sub-population increases (Fig. 3).

The BVA plot exhibits evidence of within-burst dynamics, so mpH²MM analysis of within-burst dynamics is warranted (Fig. 3, top row).

In mpH²MM analysis, the ICL-based model selection identifies the five-state model for 1 $\mu$M maltose, and the four-state model

for 1 mM maltose. Examining these models, we find that all contain a single high $S_{raw}$ state and a single low $S_{raw}$ state, with the high $S_{raw}$ state also having an $E_{raw}$ of 0, and importantly, no bursts exist in these ranges due to the use of a DCBS filter (Supplementary Tables 4 and 5). Therefore, we can conclude that these states are the result of transition in the donor and acceptor dyes for the low and high $S_{raw}$ states, respectively. The transition rates of the models and *Viterbi* analysis both show that these states are appreciably populated (Fig. 3, bottom row, Supplementary Figs. 20–22, Supplementary Table 7, and supplementary .csv file[72]), thus the use of mpH²MM analysis is vital here. Depending on the maltose concentration, the ICL of spH²MM analysis predicts different numbers of states for each concentration, and the $E_{raw}$ values of the states within these models are far less consistent (Fig. 3, vertical bars). These states are often similar to states found by mpH²MM, but their interpretation would be ambiguous if we did not have mpH²MM for additional information. In other cases, spH²MM-based states appear as a fusion of two states found by mpH²MM.

As another example of how vital mpH²MM analysis is in this case, consider the BVA signature of the apo form. Our analysis shows that the FRET dynamics for $E_{raw}$ is not due to the actual dynamics between the open conformations of MalE. This is clear since the ~ 0.7 $E_{raw}$ sub-population is not identified if maltose is not supplied. Such interpretation cannot be made from spH²MM results, due to the less consistent prediction of the number of states, and the parameters of those models. Therefore, we can confirm that MalE undergoes large-scale conformational dynamics linked to its function, mostly induced by the binding of maltose, hence it follows an induced-fit binding mechanism.

**Adapting to $\mu$sALEX: microsecond dynamics of YopO.** Finally, we demonstrate how to apply mpH²MM with $\mu$sALEX experiments. For this, we use the type-III secretion effector from *Yersinia* species, YopO[71]. We measure the conformational dynamics of a double-cysteine variant of YopO, with dyes labeling residues L113C and L497C. These labeling positions are expected to change distances upon binding to actin. Burst selection is performed using the DCBS filter, for the same reasons as in the MalE data - there are strong blinking dynamics that overwhelm the analysis otherwise. Interestingly, in the absence of actin there appears to be a single FRET sub-population in E-S plots, with tails towards dark donor and dark acceptor sub-populations. Nevertheless, BVA shows these bursts have a variance above the expected static FRET semi-circle (Fig. 4a, top panel), and hence within-burst dynamics. In the presence of bound actin (60 $\mu$M), a main sub-population is present with a shift toward lower $E_{raw}$ values, and the BVA plot suggests no signature of within-burst dynamics at that main sub-population (Fig. 4b, top panel).

Using this $\mu$sALEX data with mpH²MM, the alternation period proves to be an obstacle, causing mpH²MM analysis to fail without a key adjustment to the data. Unlike in nsALEX, multiple photons can be detected during a given alternation period of the donor or acceptor excitation lasers. This results in photons originating from donor excitation that are temporally separated from photons originating from acceptor excitation in a periodic pattern, resulting in alternating periods where no photons originating from donor excitation are detected, and alternatively periods where no photons originating from acceptor excitation are detected. When we first apply mpH²MM to $\mu$sALEX data, we find that instead of detecting states with meaningful $S_{raw}$ values, all states have $S_{raw}$ values of either 0 or 1, and transition rates are all very similar to the alternation rate, meaning that mpH²MM detects the alternation rate instead of actual conformational dynamics (Supplementary Fig. 23a, b).

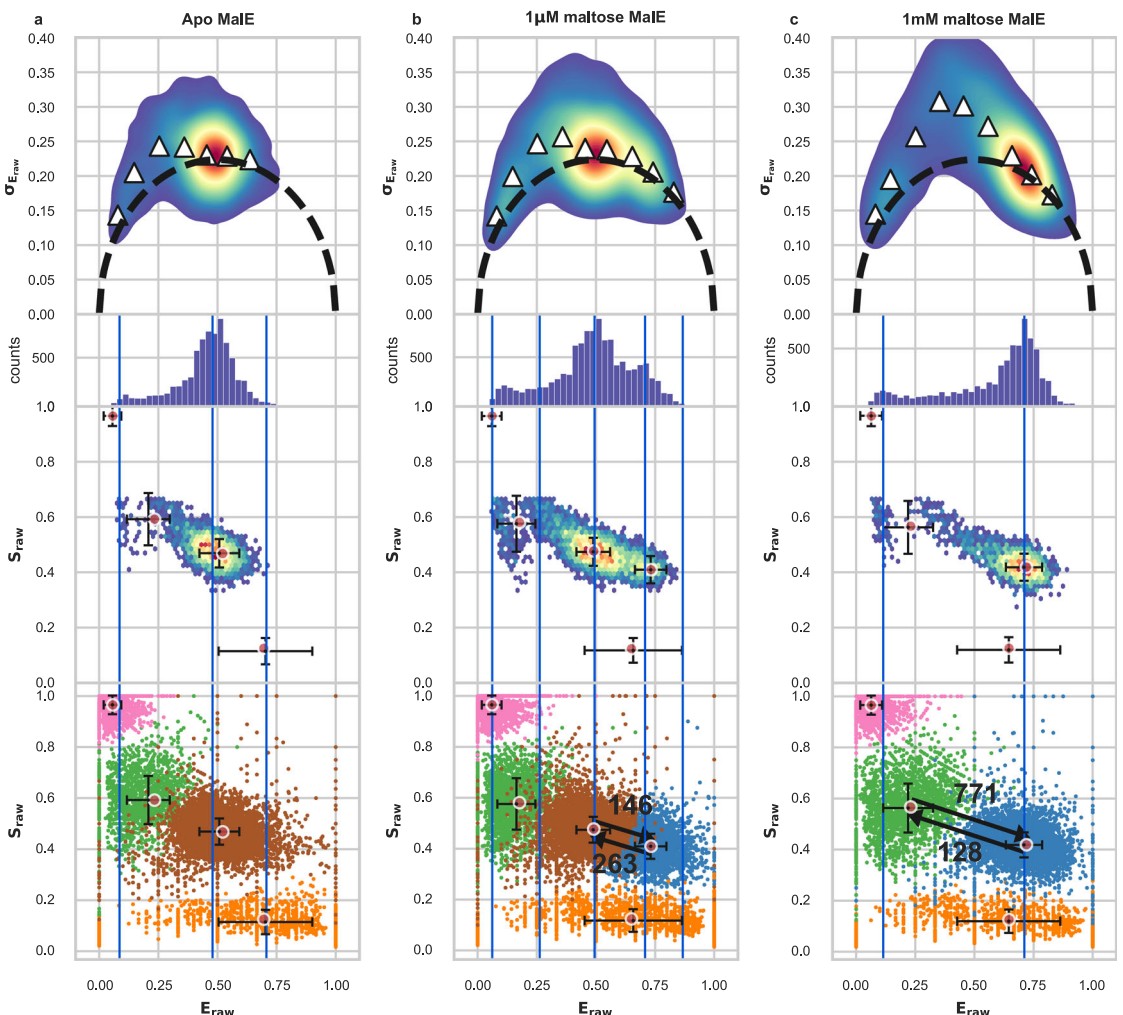

**Fig. 3 Results for MalE.** Top row: BVA of concatenated dataset. Upper middle row: $E_{raw}$ histogram of bursts. Lower middle row: E-S plot of bursts, with ICL-based selected results overlayed, red circles indicating the values derived from the ICL-based selected mpH²MM state model, and the black crosses the standard deviation of the *Viterbi*-derived dwell $E_{raw}$ and $S_{raw}$ values. Vertical blue lines represent the $E_{raw}$ values of the states from the ICL-based selected spH²MM state model. Bottom row: Dwell based E-S plots as in Fig. 2, with transition rates (in units of s$^{-1}$) between selected states indicated by arrows added. **a** apo MalE, **b** 1 μM maltose, **c** 1 m M maltose. Error bars (s.d.) for a: $n = 1556, 3925, 3042, 8665$ dwells for dark donor, dark acceptor, low FRET, and mid FRET states, respectively, for **b** $n = 2505, 7753, 4793, 9083, 4010$ dwells for dark donor, dark acceptor, low FRET, mid FRET, and high states FRET, respectively, for (**c**) $n = 1586, 5539, 4196, 7550$ dwells for dark donor, dark acceptor, low FRET, and high FRET states, respectively.

In that respect, to enable meaningful μsALEX analysis via mpH²MM that incorporates photons originating from acceptor excitation, we introduce a shift so that the times of the acceptor excitation photons overlap with the photons originating from donor excitation (see Supplementary Note 1.1.1). By doing so, the alternation period is no longer detected and meaningful dynamics with $E_{raw}$ and $S_{raw}$ values can be recovered (Fig. 4, Supplementary Fig. 23c). The usefulness of this analysis is evidenced by the detection of dark donor and dark acceptor states. Thus application of mpH²MM even to μsALEX data usually yields better results than with spH²MM. However, caution must be taken to avoid artefacts due to the alternation period. For instance, if the timescale of a transition approaches that of the alternation period, $S_{raw}$ values may be biased or averaged together due to the shift (for in-depth discussion on this topic, see Supplementary Note 1.1.2).

Applying mpH²MM to analyze the measured data of YopO in the presence of actin, the most likely model is clearly a four-state model, using an alternation period of 50 μs (20 kHz alternation rate). The ICL-based model selection identifies four states, while the BIC'-based

selection shows the four-state model to be close to the 0.005 threshold, and the five-state model can be further disregarded based on its reasonableness. Selection is more difficult for the apo results, as the two criteria disagree with ICL-based model selection that identifies three states and BIC'-based model selection that identifies five. Therefore, the most likely model is either the three-, four-, or five-state model, and examination of these models and prior knowledge of the data is necessary. The three-state model predicts states that appear as dark donor and acceptor, and a single FRET state. This model can be ruled out because the BVA shows significant dynamics around the single FRET population, and thus the single FRET state is insufficient to explain the BVA signature. The five-state model, on the other hand, suffers from the opposite problem - there are two states with very low $S_{raw}$ values, where it appears as though the dark donor state has split into two. The four-state model, however, is reasonable, showing two FRET states, dark donor and dark acceptor states (Supplementary Tables 6 and 7). Transition rates between the high and low FRET states are 12,400 s$^{-1}$ and 6,000 s$^{-1}$ for transitions from high $E_{raw}$ to low $E_{raw}$ states, and for transitions from low $E_{raw}$ to high $E_{raw}$ states, respectively. These dynamics,

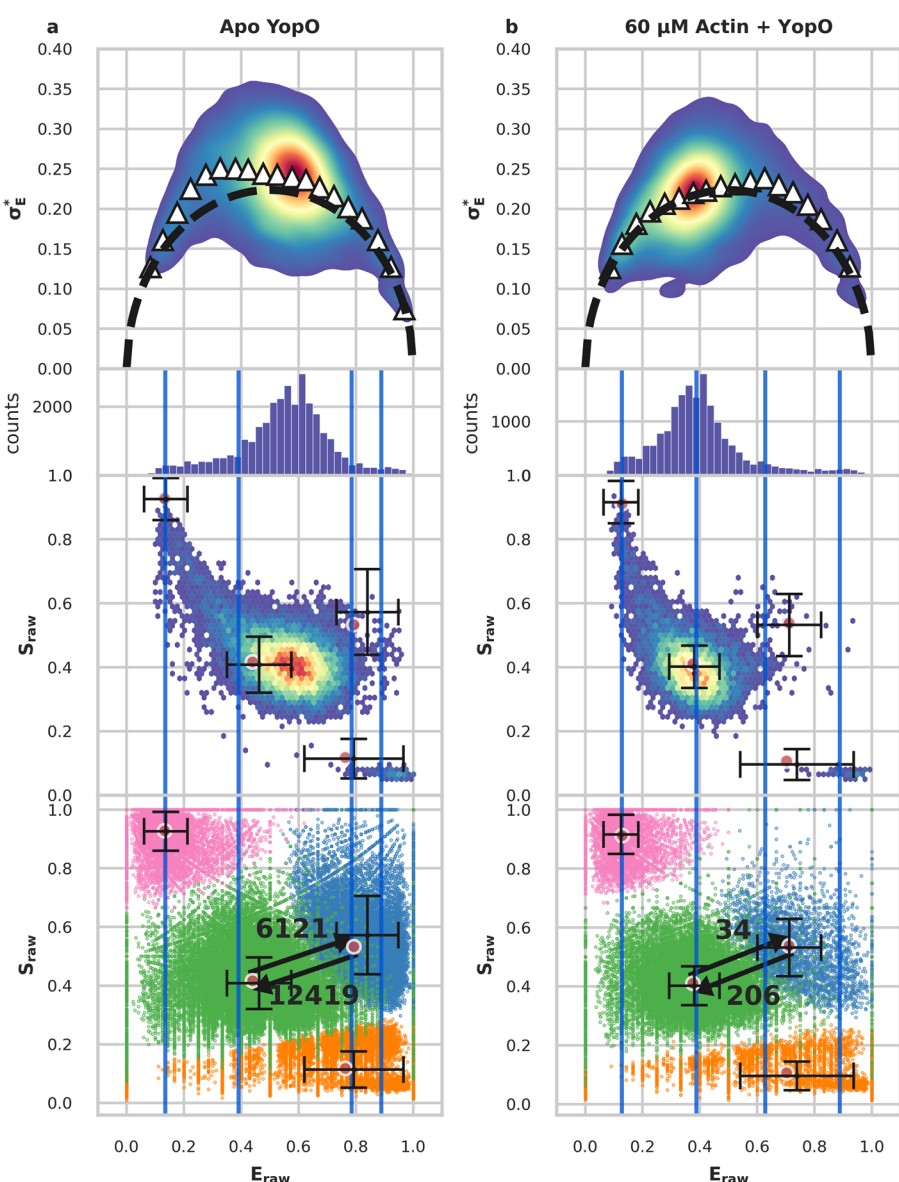

**Fig. 4 Results for YopO. a, b** Top row: BVA analysis. Upper middle row: $E_{raw}$ histogram of bursts, lower middle row: E-S plot of bursts. Red dots indicate mpH$^2$MM states. Bottom row: Dwell based E-S plots as in Fig. 2, with transition rates (in units of s$^{-1}$) between selected states indicated by arrows added. Vertical bars indicate the $E_{raw}$ values of the states for the BIC'-based spH$_{Eraw}$ model. **a** apo YopO exhibiting sub-millisecond dynamics. **b** YopO with 60 $\mu$M actin exhibiting slower within-burst dynamics, and a shift toward the lower FRET conformation. Error bars (s.d.) for **a** $n = 40,723$, 89,613, 108,641, and 89,613 dwells for dark donor, dark acceptor, low FRET, and high FRET states respectively, for **b** $n = 11,518$, 9279, 31,100, and 3105 dwells for dark donor, dark acceptor, low FRET, and high FRET states, respectively.

however, approach the timescale of the alternation rate (20 kHz; for detailed discussion and examination, see Supplementary Note 1.1.2 as well as Supplementary Figs. 24, 25). Based on the analyses of the *Viterbi*-derived dwell times, error analysis of data sub-samples, and comparison with the results of mpH$^2$MM analysis employed on other measurements using different alternation periods, we conclude that these transition rates are not artifacts, and reflect true FRET transitions in the data (see Supplementary Note 1.1.2, Supplementary Figs. 26–28 for comparison of different alternation periods, Supplementary Tables 8 and 9 for optimized model of different alternation period).

The timescale of the FRET dynamics being faster than burst duration by two orders of magnitude explains the appearance of the data in the FRET histogram as a single FRET population, yet with a signature of within-burst dynamics in the BVA plot. Inspecting the results of the mpH$^2$MM analysis, the meaning of

the within-burst FRET dynamics of YopO in the absence of actin becomes clear - it exhibits transitions in the tens of microseconds between two main FRET states intertwined with rapid transitions to dark donor and dark acceptor states. Each burst that lasts a few milliseconds contains multiple dwells in the underlying states and transitions between them, and so the bursts are averaged-out as a single main population. When comparing these results, with the analysis results of YopO in the presence of bound actin, it becomes clear that the lower $E_{raw}$ state of the two FRET states in the absence of actin is stabilized upon actin binding. Therefore, we can conclude that YopO conformational dynamics relevant to actin binding occurs intrinsically, regardless of the presence of actin, and that actin stabilizes and locks one of the pre-existing conformations.

Without using mpH$^2$MM, it would have been difficult to accurately report on this dynamics, as the FRET within-burst

dynamics is intertwined with FRET-irrelevant transitions to dark states. It should be noted that we have successfully decoupled conformational and photophysical dynamics in $\mu$sALEX data without the use of fluorescence lifetimes.

## Discussion

MpH$^2$MM increases both the information content of the results and the sensitivity of the H$^2$MM algorithm to differences in the photon streams that are too subtle when examining only a single parameter. We have shown that mpH$^2$MM is able to disentangle dark acceptor states from low FRET states that have structural meaning. We have exhibited the advantage of using mpH$^2$MM to elucidate an accurate quantitative picture on two proteins with two types of conformational dynamics that serve their function: (1) MalE with conformational dynamics induced by maltose binding, and (2) YopO with conformational dynamics occurring intrinsically, with actin binding stabilizing one of the states. In both cases, the overall picture is complicated by having the FRET-relevant transitions intertwined with the FRET-irrelevant dye transition to dark states, and not taking these into account could result in wrongly elucidated quantities and potentially wrong interpretations. Of note is the rapid conformational dynamics on the order of tens of microseconds in YopO when actin was absent. The exact description of the dynamics was possible using mpH$^2$MM on $\mu$sALEX, and hence did not necessarily require analysis of the correlation of donor fluorescence lifetimes with ratiometric FRET values, as can be done using FRET lines fits to E-$\tau_D$ 2D plots in lifetime-based smFRET[44]. As $\mu$sALEX and nsALEX setups are now commonly used, the acceptor excitation stream is usually available, therefore, mpH$^2$MM maximizes the use of available data for characterizing rapidly interconverting sub-populations.

MpH$^2$MM is, therefore, a powerful tool for the quantification of rapid conformational dynamics in a variety of systems, while also extracting information that can be used to extract inter-dye distance distributions. The integration of the acceptor excitation photon stream is critical in this process, as we have shown that spH$^2$MM often conflates photophysical and conformational states, leading to incorrect $E_{raw}$ and transition rate constants. Comparing a given protein or other biomacromolecular system with different ligands, or concentrations of ligands, it is possible to discriminate when a system demonstrates intrinsic conformational dynamics or conformational changes triggered by ligands. MpH$^2$MM provides accurate quantitative measures of both transition rates and mean $E_{raw}$ values, the latter of which can be converted into accurate mean FRET efficiency values with the proper correction factors for the system[79]. Such information can then be converted into mean inter-dye distances, which provide invaluable information for FRET-based integrative structural models[7,33,79].

The success of integrating the acceptor excitation stream into our analysis, suggests that a similar approach could also be employed in camera-based smFRET applications. Alternating laser excitation, and HMM algorithms are commonly employed in analysis of data, although the information of the acceptor excitation stream is discarded in these analyses, and only used for truncating trajectories upon acceptor bleaching. The present mpH$^2$MM algorithm is inappropriate for such data as the information is based on intensity and a constant frame rate, instead of single photon arrivals with variable interphoton times. Alternatively, the introduction of SPAD arrays[80] should allow analysis of immobilized molecules with single photon precision, which would allow for analysis of such data directly using mpH$^2$MM.

mpH$^2$MM is also not restricted to our demonstrated application in two detector setups with nsALEX and $\mu$sALEX. The most obvious application of mpH$^2$MM beyond ALEX, is with the multiple photon streams in multi-parameter fluorescence detection (MFD)[34,43], or with multi-color smFRET-based measurements[62–69]. Here, three or even four spectrally-distinct dyes are attached to the biomolecule of interest, and each produces a distinct photon stream. This enables the simultaneous observation of multiple inter-dye FRET efficiencies at once. If qualitative tests indicate that such a system is undergoing within-burst dynamics, mpH$^2$MM is well-suited to extract the transfer efficiencies relevant to the underlying dynamically interconverting sub-populations. Applying these methods is as simple as assigning an index to each photon stream. We include a supplementary Jupyter notebook using a developer version of FRETBursts[81] that accepts fluorescence anisotropy information from multi-parameter fluorescence detection, or from MFD coupled to pulsed interleaved excitation[34,43], and demonstrate mpH$^2$MM's ability to disentangle fluorescence anisotropies on data kindly provided by Cao et al.[82]. Values within the emission probability matrix can then be used as intensities to calculate all relevant ratiometric values. Multiple conformational sub-populations interconverting at sub-millisecond timescales could be simultaneously measured and disentangled with such a setup. Information on fluorescence anisotropy could also be incorporated, which, depending on the labeling scheme could report on dye steric restriction or oligomeric state of the system in question.

In this work, we used two ratiometric parameters drawn from ratios of photon counts of the photon streams available in ALEX-based measurements within the mpH$^2$MM framework. In some smFRET measurements, such as in nsALEX, the photon nanotimes, which are the basis for fluorescence lifetime data, can also be considered as a parameter within the mpH$^2$MM framework. However, unlike $E_{raw}$ and $S_{raw}$, which are approximately binomially distributed, photon nanotimes distribute exponentially or sometimes according to a sum of exponentials. To transform photon nanotime data into a parameter that is also centrally distributed, and hence one that can be used within the mpH$^2$MM framework, we propose a method for mapping the non-binomially distributed lifetime to a binomially distributed parameter amenable to mpH$^2$MM (see Supplementary Note 3 for further details).

The new H2MM_C python package makes H$^2$MM analysis much more practical, most analysis, for up to six states, take less time than the data acquisition times, given our modest hardware (a 2 year old middle-tier gaming laptop). See Supplementary Note 4 and Supplementary Tables 10 and 11 for system requirements and the duration of calculations in this paper. The supplied Jupyter notebooks provide examples for how to execute mpH$^2$MM using FRETBursts. Experimenters using other platforms must utilize their knowledge of the fine details of their data to properly filter and cast their data into the simple and general format that the H2MM_C package[76] accepts. We also provide an in-depth tutorial available on Zenodo[83].

## Methods

**Production of YopO and MalE variants**. The double-cysteine variant YopO L113C/L497C is produced and purified as described in Peter et al.[71] and kindly provided by Gregor Hagelüken and Martin Peter, Institute of Structural Biology (University of Bonn). The double-cysteine variant MalE T36C/S352C is generated and purified according to methods reported previously[17].

**Labeling of MalE**. The MalE variant T36C/352C is stochastically-labeled with Alexa Flour™ 555 and Alexa Fluor™ 647 dye derivatives as described in Peter et al. and deBoer et al.[17,84]. The His$_6$-MalE double variant (200 $\mu$g) is incubated with 1 mM DTT and loaded immediately after on 200 $\mu$L (wet volume) Ni-Sepharose 6 Fast Flow resin, pre-equilibrated with labeling buffer 1 (50 mM Tris-HCl pH 7.4, 50 mM KCl). After a washing step with 50 column volumes labeling buffer 1, the loaded resin is incubated overnight at 4 °C with 5-fold excess (25 nmol of each fluorophore dissolved in 1 mL of labeling buffer 1. Next, the resin is further washed

with 50 column volumes labeling buffer 1 to remove the excess unbound fluor-ophores. Labeled protein is eluted with 800 $\mu$L elution buffer (50 mM Tris-HCl pH 8.0, 50 mM KCl, 500 mM imidazole) and further purified by size-exclusion chromatography (ÄKTA pure system, Superdex 75 Increase 10/300 GL column, GE Healthcare). Protein concentration is determined using the protein extinction coefficient and corrected for direct absorption of the fluorophores at 280 nm. Labeling efficiencies are estimated to be at least 60% for each fluorophore individually and donor-acceptor pairing at least 20%.

Labeled MalE is stored in 50 mM Tris-HCl pH7.4, 50 mM KCl and 1 mgmL$^{-1}$ bovine serum albumin (BSA) at 4 °C for no more than 3 days. Concentrations ranged between 10 to 100 nM.

**Labeling of YopO**. The protein variant YopO L113C/L497C is stochastically-labeled with fluorophore-linked maleimide derivatives, as described previously[84]. Briefly, 200 $\mu$g of protein is incubated with 5 mM DTT at 4 °C for 30 min, to prevent oxidation of the cysteine thiol groups. The protein is loaded onto a PD Mini-Trap G-25 column (GE Healthcare) pre-equilibrated with Buffer A (50 mM Tris-HCl pH 7.4, 50 mM KCl) and subsequently eluted with 1 mL of Buffer A by gravity gel filtration, in order to eliminate the excess of DTT. The eluted protein is incubated overnight at 4 °C with 50 nmol, respectively, of Alexa Fluor™ 555- and Alexa Fluor™ 647- C$_2$ maleimide (ThermoFisher Scientific). Excess dyes are removed again by gravity gel filtration using a PD Min-Trap G-25 column, as described above. The labeled protein is further purified from residual dyes and soluble aggregates by size-exclusion chromatography (SEC), with a Superdex™ 75 Increase 10/300 GL column, on an ÄKTA pure system (GE Healthcare). Protein concentration is determined using the protein extinction coefficient and corrected for direct absorption of the fluorophores at 280 nm.

Labeling efficiencies are estimated to be at least 60% for each fluorophore individually and donor-acceptor pairing at least 20%.

**Experimental setup**

*Experimental setup for studies of HP3.* We performed the nsALEX smFRET measurements of the doubly-labeled DNA hairpin construct[70] in the presence of 50, 100, 200, 250, 300, and 350 mM sodium chloride, using a confocal-based setup (ISS™, USA) assembled on top of an Olympus IX73 inverted microscope stand. We use a pulsed picosecond fiber laser ($\lambda$ = 532 nm, pulse width of 100 ps FWHM, operating at 20 MHz repetition rate and 100 $\mu$W measured at the back aperture of the objective lens) for exciting the Cy3B donor dye (FL-532-PICO, CNI, China), and a pulsed picosecond diode laser ($\lambda$ = 642 nm, pulse width of 100 ps FWHM, operating at 20 MHz repetition rate and 60 $\mu$W measured at the back aperture of the objective lens) for exciting the ATTO 647N acceptor dye (QuixX® 642-140 PS, Omicron, GmbH), delayed by 25 ns. The laser beams pass through a polarization maintaining optical fiber and then further shaped by a linear polarizer and a halfwave plate. A dichroic beam splitter with high reflectivity at 532 and 640 nm (ZT532/640rpc, Chroma, USA) reflects the light through the optical path to a high numerical aperture (NA) super apochromatic objective (60X, NA = 1.2, water immersion, Olympus, Japan), which focuses the light onto a small confocal volume. The microscope collects the fluorescence from the excited molecules through the same objective, and focuses it with an achromatic lens (f = 100 mm) onto a 100 $\mu$m diameter pinhole (variable pinhole, motorized, tunable from 20 $\mu$m to 1 mm), and then re-collimates it with an achromatic lens (f = 100 mm). Then, donor and acceptor fluorescence are split between two detection channels using a dichroic mirror with a cutoff wavelength at $\lambda$ = 652 nm (FF652-Di01-25x36, Semrock Rochester NY, USA). We further filter the donor and acceptor fluorescence from other light sources 585/40 nm (FF01-585/40-25, Semrock Rochester NY, USA) and 698/70 nm (FF01-698/70-25, Semrock Rochester NY, USA) band-pass filters, respectively, and detect the donor and acceptor fluorescence signals using two hybrid photomultipliers (Model R10467U-40, Hamamatsu, Japan), routed through a 4-to-1 router to a time-correlated single photon counting (TCSPC) module (SPC-150, Becker & Hickl, GmbH) as its START signal (the STOP signal is routed from the laser controller). We perform data acquisition using the VistaVision software (version 4.2.095, 64-bit, ISS™, USA) in the time-tagged time-resolved (TTTR) file format. After acquiring the data, we transform it into the photon HDF5 file format[85] for easy dissemination of raw data to the public, and easy input in the FRETBursts analysis software.

*Experimental setup for studies of MalE.* The nsALEX measurements on MalE are performed using a home-built setup, assembled around an Olympus IX73 inverted microscope stand. We use a picosecond pulsed diode laser ($\lambda$ = 532 nm, pulse width of 100 ps FWHM, operating at 20 MHz repetition rate and 32 $\mu$W at the back aperture of the objective) for exciting the Alexa Fluor™ 555 donor (LDH-P-FA-530B, Picoquant GmbH), and a picosecond pulsed diode laser ($\lambda$ = 640 nm, pulse width of 90 ps FWHM, operating at 20 MHz repetition rate, and 20 $\mu$W at the back aperture of the objective) to excite the Alexa Fluor™ 647 acceptor (LDH-D-C-640, Picoquant, GmbH), driven by the same PDL828 "Sepia II" (Picoquant, GmbH) controller. The laser light is guided into the microscope by a dual-edge beamsplitter (ZT532/640rpc Chroma/AHF, GmbH) and focused to a diffraction-limited excitation spot by an oil immersion objective (UPLSAPO 60XO, Olypus). The emitted light is collected through the same objective, spatially filtered through a 50 $\mu$m pinhole, and spectrally split into donor and acceptor channels by a single-edge

dichroic mirror (H643 LPXR, AHF). The emission is filtered (donor: BrightLine HC 582/75, Semrock/AHF, acceptor: Longpass 647 LP Edge Basic, Semrock/AHF) and the signal is recorded with avalanche photodiodes (SPCM-AQRH-34, Excelitas) and a TCSPC module (HydraHarp400, Picoquant, GmbH). Data was acquired with Picoquant SymPhoTime 64 v2.7.

Coverslips are passivated with 1 mg mL$^{-1}$ BSA in PBS buffer before adding around 100 $\mu$L of sample. MalE stock solution is diluted to ~ 50 pM concentration in 50 mM Tris-HCl pH 7.4, 50 mM KCl, and either, none, 1 $\mu$M or 1 mM of the ligand maltose.

*Experimental setup for studies of YopO.* The $\mu$sALEX measurements of YopO are performed using the setup in Gebhardt et al.[86]. These are conducted on the same home-built microscope as the MalE experiments, built around an Olympus IX71 base, although the lasers and dichroics are replaced as described below. We use a continuous wave $\lambda$ = 532 nm diode laser (OBIS 532-10-LS, Coherent, USA) laser with 60 $\mu$W power measured at the back aperture of the objective to excite the donor Alexa Fluor™ 555 dye, and a continuous wave $\lambda$ = 640 nm diode laser (OBIS 640-100-LX, Coherent, USA) with 25 $\mu$W power measured at the back aperture of the objective. The lasers are distally modulated by TTL pulses with an alternating frequency of 10 kHz, 20 kHz, and 100 kHz, for an alternation period of 100 $\mu$s 50 $\mu$s, and 10 $\mu$s, respectively. The lasers are combined and coupled into a polarization maintaining single-mode patch cable (P-3-488PM-FC2, Thorelabs, USA). The laser light is reflected into the objective by a dual-edge dichroic mirror (ZT532/640rpc, Chroma/AHF) and focused by a water immersion objective (UPlanSApo 60/1.2w, Olympus, GmbH). The dichroic mirrors, fluorescent filters and avalanche photodiodes are identical to those used for acquisiton of MalE data.

Coverlips are passivated with BSA as in MalE measurements. 100 $\mu$L of YopO solution, diluted to between 50 pM and 80 pM is used for each measurement in 50 mM Tris-HCl pH 7.4, 50 mM KCl. For measurements with actin, the buffer also contained 50 $\mu$M non-muscle human actin protein (Cytoskeleton, Inc) and 0.2 mM ATP and 0.2 mM CaCl$_2$.

Data is acquired using labVIEW v7.1 software as presented in Ingargiola et al.[87].

**Burst selection**. All data processing and analysis is performed using Jupyter Notebooks available in supplementary dataset, along with the accompanying photon-HDF5 files containing the raw data[72]. We perform burst search and selection using the FRETBursts analysis software[88]. The background is assessed per each 30 s of acquisition, and bursts are identified as time periods were the instantaneous photon count rate of a sliding window of $m = 10$ consecutive photons is at least $F = 6$ times higher than the background rate. Bursts in the normal selection are selected if they include at least 30 photons in total between all streams. Visualizations are performed using FRETBursts' dplot function, or mat-plotlib when greater customization is desired.

**Single and multi-parameter H$^2$MM analysis**. Bursts identified by FRETBursts are then converted into a format readable by the H2MM_C software[76], by a simple function supplied in the Jupyter notebooks available in supplementary dataset[72], this function is also responsible for applying the shift to acceptor excitation photons in $\mu$sALEX experiments (Supplementary Note 1.1.1). In spH$^2$MM, only photons arising from donor excitation are considered, assigned to either donor or acceptor streams, identified by index 0 or 1, respectively, depending on at which detector they arrived. MpH$^2$MM also considers photons arriving during acceptor excitation, assigning these photons an index of 2. All H$^2$MM calculations are performed within the Jupyter notebooks, available in supplementary dataset[72], using the Python package by Paul David Harris[76]. We use the H$^2$MM algorithm (both single- and multi-parameter) to test how well different state models describe the data.

*Model selection.* To choose the best model, we primarily use the ICL[74,75], where the state model reaching a minimal ICL is generally considered the one that describes the data best, with minimal free parameters. We always calculate sufficient numbers of state models to ensure ICL is minimized. The ICL parameter is defined in Eq. (1):

$$ICL(m) = -2\ln(\mathbf{p}(\mathbf{y}, \hat{\mathbf{s}}|m, \hat{\lambda}_m)) + K\ln(n) \quad (1)$$

where $\ln(\mathbf{p}(\mathbf{y}, \hat{\mathbf{s}}|m, \hat{\lambda}_m))$ is the posterior probability of the most likely state path, as determined by the *Viterbi* algorithm, $K$ is the number of free parameters in the model, and $n$ is the number of photons in all bursts in the data set. $K$ is calculated as in Eq. (2):

$$K = q^2 + (r - 1)q - 1 \quad (2)$$

where $q$ is the number of states the state model represents, and $r$ is the number of photon streams used for the calculation of all of the parameters that are assessed. For spH$^2$MM, $r = 2$, while for nsALEX mpH$^2$MM, $r = 3$. The ICL is preferable as an extremum-based criterion over the previously proposed threshold based on the modified Bayes Information Criterion (BIC')[7]. See supplementary dataset[72] for Jupyter notebooks testing the reliability of ICL with simulated data sets generated using PyBroMo[89] (https://github.com/OpenSMFS/PyBroMo/releases/tag/0.8.1; was

utilized in previous works[7,85,90]). We use the *Viterbi* algorithm to find the most likely state path based on the posterior probability.

*Viterbi analysis.* From the state path, photons are separated into dwells, each of which can be assigned a duration, a mean $E_{raw}$, and for mpH$^2$MM, a mean $S_{raw}$. This also allows bursts to be classified by which and how many states are present. As one measure of error, we use the weighted standard deviation and the weighted standard error of the $E_{raw}$ and $S_{raw}$ as a proxy for the standard error of the H$^2$MM model (see Supplementary Note 1.3 for full derivation).

*Error analysis by variance of subsets.* Analysis of the variance of subsets is another method to assess the error of parameters (see Supplementary Note 1.4 for detailed description). This is implemented as a function in the Jupyter notebooks in the supplementary dataset[72]. This is an attractive approach, as it does not depend on any most likely state path like in the *Vieterbi* based approach. This method, however, is significantly more computationally expensive than the *Viterbi* approach.

**Reporting summary.** Further information on research design is available in the Nature Research Reporting Summary linked to this article.

## Data availability

The photon-HDF5 data, Jupyter notebooks, .csv data and H2MM_C code that support the findings of this study are available in the Zenodo and github repositories with the identifiers at: https://doi.org/10.5281/zenodo.5566809, and https://doi.org/10.5281/zenodo.5535302[72,76].

## Code availability

The H2MM_C library used in this study is available on github https://github.com/harripd/H2MMpythonlib (commit 1f3d0a84f149d21a740161372526eb3742027602). The FRETbursts used in this study is available on github https://github.com/harripd/FRETBursts (commit 315c60d3791aa93cf2ec6e880003174c8192fc88). The phconvert code used n this study is available on github https://github.com/Photon-HDF5/phconvert v0.9 (commit 3a86e58f11f77e21c2a02a1d9453060db6811c9c). The PyBroMo code used in this study is available on github: https://github.com/tritemio/PyBroMo v0.8.1 (commit 8403ae750ff68796ef4118dd497478cf54355382). labVIEW code is available on github: https://github.com/multispot-software/MultichannelTimestamper.

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

## Acknowledgements

We thank Gregor Hagelücken and Martin Peter from the Institute of Structural Biology (University of Bonn, GER) for providing YopO. We would like to thank Robert Quast and Emmanuel Margeat for insightful discussions regarding the implementation of mpH$^2$MM for the analysis of 4-detector nsALEX measurements (2-color smFRET, with fluorescence anisotropies), based on their existing data[82]. We would also like to thank Demain Lieberman for his helpful discussion regarding implementation of H$^2$MM code, and Bill Harris for his help in enabling the H2MM_C code to work on Windows and Linux. This paper was supported by the National Institutes of Health (NIH, grant R01 GM130942 to S.W. and E.L. as a subaward), the National Science Foundation (NSF, grants 1818147 and 1842951 to S.W.), the Human Frontiers Science Program (HFSP, grant RGP0061/2019 to S.W.), the Israel Science Foundation (ISF, grant 3565/20 to E.L., within the *KillCorona* – Curbing Coronavirus Research Program), the Milner Fund (to E.L.), and the Hebrew University of Jerusalem (start-up funds to E.L.). Work in the lab of T.C. was financed by Deutsche Forschungsgemeinschaft (SFB863, project A13 and GRK2062, project C03), an ERC Starting Grant (No. 638536 – SM-IMPORT to T.C.) and by the Center of Nanoscience Munich (CeNS).

## Author contributions

E.L. performed HP3 nsALEX measurements. C.G. performed MalE nsALEX measurements. A.N. and T.C. performed YopO μsALEX experiments. P.D.H. analyzed data and

contributed analytical tools. P.D.H. & E.L. designed and performed the research and composed the initial manuscript. P.D.H., A.N., C.G., T.C., S.W. & E.L. discussed the data and contributed to the final version of the manuscript.

## Competing interests

The authors declare no competing interests.
