## [Peer Review File · Nature Communications]

Reviewers' Comments:

Reviewer #1:

Remarks to the Author:

The manuscript "Multi parameter photon by photon hidden Markov modeling" by Harris et al. follows a hidden Markov modeling approach in the analysis of photon trajectories acquired from 2c-nsALEX experiments to infer the model parameters including: number of subpopulations, mean proximity ratios (PR), stoichiometry ($S_{\{PR\}}$), the FRET related transitions and FRET unrelated transitions due to photo physical features of the attached dyes. The framework reported in this manuscript is built off of the existing H2MM approach presented in the work of Pirchi et al.

<https://pubs.acs.org/doi/abs/10.1021/acs.jpcc.6b10726> where the framework was carried out only for the donor excitation resulting photon trajectories in both donor and acceptor channels. Here, instead, the authors incorporate the analysis of photon trajectory due to acceptor excitation in their analysis. Thus, unused photon trajectory in the previous studies, already present in ALEX and PIE setups, is now used to extract more information for the FRET dynamics of the probed single molecule. Notably, the work addresses some of the shortcomings of existing single photon trajectory analysis from ALEX and PIE setups regarding the difficulty of differentiating between subpopulations when they share the same FRET efficiencies (PRs) for example subpopulations of low FRET population and blinking acceptor due to acceptor excitation photon trajectory. Furthermore, the approach, results and interpretation are scientifically sound; and the manuscript is well written. Nonetheless, I have several major concerns about this work and manuscript that I describe further details below and that do not allow me to recommend for publication at least not in its current form for Nature Communications.

Specific concerns to be addressed:

More important:

1. The work sounds like it is a fundamentally new method although it is really adding one more photon trajectory analysis in the existing framework of H2MM by Pirchi et al. <https://pubs.acs.org/doi/abs/10.1021/acs.jpcc.6b10726>. Therefore, the novelty of the method is in question.
2. In Sottini et al. <https://www.nature.com/articles/s41467-020-18859-x> they also use a similar analysis to differentiate between subpopulations based on Viterbi paths where they also consider both ALEX and PIE setups. In this case, how does the current method presented in this manuscript provide more information or do so more robustly than the existing methods?
3. The mapping of E-S plots for PR and $S_{\{PR\}}$ relies purely on the Viterbi paths obtained based on the assumed number of subpopulations and the estimated parameters. Therefore, it is not clear how those error bars are obtained from Viterbi paths. Furthermore, error bars can only incorporate information about the assumed model. Therefore, it is not clear how the model uncertainty is incorporated in the estimates.
4. In previous work presented in Lerner et al., "Transcription bubble of RNAP-.." , it was shown that the FRET dynamics are governed by 4 subpopulations. Here, in this work, they seem to recover the same subpopulations using the $S_{\{PR\}}$ values estimated from their Viterbi paths. Therefore, it is not clear what the new method fundamentally adds.

Less important:

5. The determination of the number of subpopulations relies in integrated complete likelihood (ICL) information. In other words, given the maximum likelihood estimates of the model parameters, the model is determined. However, this method does not propagate uncertainty on the possible models from the potential parameter estimation. Therefore, it is not clear how the error bounds are created for the estimates.
6. In all figures' panel (b), the y-axis label does not seem to be compiled successfully.

7. In the end of "RNAP promoter open complex" section, the authors mention the use of fluo-rescent lifetimes to differentiate between the increase or decrease of the quantum yields of acceptor and donor dyes, respectively. Is it possible for the authors to explain how they extract these lifetimes from the experiments for non-experts in the field, at least a sentence stating that it is attributed to the pulse excitation setup?
8. The authors directly start the analysis of experimental data however, I suggest that they showcase their method on simulated data where the plain H2MM fails. This would help in easily seeing the difference of their method with the existing competing methods.
9. On page 2, the authors already introduced PR for proximity ratio and on page 3, they prefer to go back to "proximity ratio". I suggest them to stick to their convention consistently.
10. Can you provide inline equation for $S_{\{PR\}}$?

Reviewer #2:

Remarks to the Author:

Summary

Harris et al. introduce multi-parameter H2MM (mpH2MM) and provide 'proof of concept' on smFRET data using a DNA hairpin and the transcription bubble formed with RNA Polymerase (RPO). They propose the use of the integrated complete likelihood (ICL) to determine the most likely state-model. Using the new mpH2MM they are able to additionally discriminate dark acceptor and donor sub-populations in a DNA hairpin. For the transcription bubble data, analysis using mpH2MM additionally enables the recovery of stoichiometry information. While the paper presents major advances in the speed and functionality of photon-by-photon H2MM analyses for smFRET, there are a few issues to address before publication in Nature Communications can be recommended.

Detailed overview

Multi-parameter H2MM as presented here allows access to a further dimension for analysis of biomolecule dynamics using smFRET. The authors' description of their implementation of the H2MM code in C as 'dramatically faster' undersells the improvement – the implementation completely changes the ability to perform H2MM analysis on large/complex datasets. While this improvement is game-changing, the utility of mpH2MM over 'spH2MM' is not well demonstrated. While the discrimination of dark acceptor/donor sub-populations for the DNA hairpin perhaps sheds some light on 'photo-physical transitions' this is not relevant from a biomolecular perspective and was not used to provide any new information about the system's dynamics. Similarly, for the RPO, while the authors present an avenue to elucidate the underlying photo-physical mechanism causing a decreased SPR for one of the sub-populations, the proposed use of this additional information to model the structure is not demonstrated to be necessary or effective.

Major revisions

1) mpH2MM as implemented here is extremely powerful and likely to supplant current smFRET dynamics analysis methods. While the use of multi-parameter analysis has been shown to elucidate some further photo-physical information its usefulness for 'conformational transitions' was not well supported. The use of two previously well-characterized systems limits the potential impact of this work. In the conclusions it is stated that mpH2MM increases the 'information content of the results and the sensitivity of the H2MM algorithm' – a discussion about the latter comparing the results derived from spH2MM and mpH2MM is missing. Providing this is strongly recommended.

For example, a key question the authors need to address in the discussion is:

For DNA hairpin if BVA shows there is no within-burst FRET dynamics what is the point of the 'dark

acceptor' state? Couldn't this be filtered out using a dual colour burst search (DCBS)? If a DCBS is used (or a minimum AexAem threshold) is mpH2MM any better than spH2MM at determining the 'biologically relevant' states?

2) While we agree in principle with the authors' own suggestion that the 'most obvious application' of mpH2MM being in multi-color smFRET measurements, this is not demonstrated. The considerable commitments required to perform these experiments is appreciated and we would not require these for publication, however, a demonstration of mpH2MM's utility using simulated data would be greatly add to the impact of the manuscript.

3) For this work to be impactful it needs to be accessible and usable by non-computer scientists and consequently we make the following comments: the Jupyter notebooks as uploaded have runtime errors from the authors – they should show complete successful outputs from running in their hands. The 'Multi-parameter photon-byphoton [sic] hidden Markov modeling dataset' (<https://doi.org/10.5281/zenodo.4671393>) is uploaded as a series of files - we strongly recommend re-uploading to maintain the directory as the notebooks currently reference a non-existent hierarchy. Also, it is not possible to proceed through the notebooks as some steps reference inputs generated with different names to those output in previous steps thereby raising errors and the code cannot run without modification. Where a name change is required to match outputs this should be commented in the code. The notebooks in the second repository 'harrisd/H2MMpythonlib: H2MM_C' (<https://doi.org/10.5281/zenodo.4671440>) do not use the C implementation of the H2MM code and cannot run as the non-C H2MM module (py script) is replaced by the C version. After some modifications to the notebooks we were able to run the H2MM_C and as mentioned it represents a dramatic improvement in the speed of analysis. However, we have not been able to proceed beyond the 'viterbi_sort' method which repeatedly kills the kernel. Ideally everything should be uploaded in a way that others can run, including a list of minimum hardware requirements (if any) as well as details on the python environment and dependencies - as we were unable to perform the full analysis in our hands.

Minor revisions:

The authors present the BIC and ICL as discriminators for the number of states. Can they please comment on how much of difference in the ICL is needed to confirm adding the extra state is significantly better? Looking at Figures 2C and 3C this is not immediately obvious.

The authors need to be clear as to exactly how the plots in Fig 2e,f and 3e,f are generated, if any burst search is applied etc. This additional detail could appear in the figure legend or the Materials and Methods.

Why will the new method not work for microsecond ALEX? How much extra effort would it be to make this work? Such an expansion would be extremely impactful increasing the reach of the new analysis to a much wider userbase who don't use pulsed lasers.

'occurring at times from picoseconds to seconds and slower[3].'

- Should state rates rather than times if saying 'slower' or changed to longer

'This is indeed the case in the coupling of catalysis with substrate binding domain dynamics in some enzymes[4, 5], the dynamics of the DNA bubble in transcription initiation to support transcription start site selection[6, 7], involved in DNA mismatch repair[8], to facilitate protein translocation across membranes[9], assisting chaperone action[10], involved in the allosteric regulation of the AAA+ disaggregase[11], influences the selectivity in ABC importers[12], and many other important bio-processes, in which structural dynamics is coupled to biological function or influences it[1,2].'

- This single sentence needs re-writing for clarity probably into several shorter sentences.

'Nevertheless, while advanced smFRET setups often detect multiple parameters beyond the simple proximity ratio, such as in alternating-laser excitation (ALEX; also known as pulsed-interleaved excitation, PIE)[28, 29] or in multi-color smFRET-based measurements[30–37], H2MM in its current iteration only reports on the proximity ratio of a single donor-acceptor pair of dyes. Here, we introduce multi-parameter H2MM (mpH2MM), which enables incorporation of multiple parameters, through additional photon streams in nanosecond ALEX (nsALEX)(a.k.a. PIE)[28, 29] experiments.'

- Referencing both ALEX and nsALEX as 'a.k.a. PIE' is confusing and can be misleading to broad readership of the journal. Be clear about the nomenclature here - nsALEX is indeed PIE, but ALEX can be microsecond ALEX (using continuous wave lasers) so is not necessarily PIE (which always uses shorter pulses).

'However, after using mpH2MM, it turns out that the four-state model is the state-model that best explains all of the data, including the acceptor-excitation data'

- 'it turns out that' reads too informally – reword or can remove transition altogether

'Using two-color microsecond ALEX experiments, Robb et. al. has shown'

- Suggest rewording to 'have shown' per conventional usage

'Multi-parameter H2MM increases both the information content of the results, and the sensitivity of the H2MM algorithm to differences in the photon streams'

- photon streams

As ALEX or PIE setups are now ubiquitous

- Not sure about the term ubiquitous here.

'Conversion of the emission probability matrix into useful $PR_{i,j}$ values'

- No previous mention or declaration of what $PR_{i,j}$ values are

'Experimenters using other platforms must utilize their knowledge of the peculiarities of their data'

- Suggest changing 'peculiarities' to particulars

'We use the H2MM algorithm (both single- and multi-parameter) to test how well 2, 3 or 4 state-models describe the photon data'

- 2 is subscript rather than superscript

This also allows bursts to be classified by which and how many states are present.

- bursts to be classified

Various spelling/typographic errors in the notebooks.

REVIEWER COMMENTS

Reviewer #1 (Remarks to the Author):

The manuscript “Multi parameter photon by photon hidden Markov modeling” by Harris et al. follows a hidden Markov modeling approach in the analysis of photon trajectories acquired from 2c-nsALEX experiments to infer the model parameters including: number of subpopulations, mean proximity ratios (PR), stoichiometry ($S_{\{PR\}}$), the FRET related transitions and FRET unrelated transitions due to photo physical features of the attached dyes. The frame-work reported in this manuscript is built off of the existing H2MM approach presented in the work of Pirchi et al. <https://pubs.acs.org/doi/abs/> where the framework was carried out only for the donor excitation resulting photon trajectories in both donor and acceptor channels. Here, instead, the authors incorporate the analysis of photon trajectory due to acceptor excitation in their analysis. Thus, unused photon trajectory in the previous studies, already present in ALEX and PIE setups, is now used to extract more information for the FRET dynamics of the probed single molecule.

Notably, the work addresses some of the shortcomings of existing single photon trajectory analysis from ALEX and PIE setups regarding the difficulty of differentiating between subpopulations when they share the same FRET efficiencies (PRs) for example subpopulations of low FRET population and blinking acceptor due to acceptor excitation photon trajectory. Further-more, the approach, results and interpretation are scientifically sound; and the manuscript is well written.

We would like to thank reviewer #1 for his positive view on our work.

Nonetheless, I have several major concerns about this work and manuscript that I describe further details below and that do not allow me to recommend for publication at least not in its current form for Nature Communications.

Dear reviewer #1.

Following your in-depth analysis of our work, we have added additional developments as well as experimental details, to fully answer the concerns you and reviewer #2 raised. These include:

1. We have added simulations to validate mpH²MM, and two additional biomolecular systems (maltose-binding protein, MalE, and actin-binding protein, YopO)
2. With the additional data we have demonstrated the ability to distinguish between different binding mechanisms
3. We have introduced additional ALEX-based methods to allow mpH²MM to be used with μ sALEX setups
4. We have clarified the text to highlight how photon streams are directly incorporated into the MLE approach, prior to any use of the *Viterbi* algorithm

We also note that upon inclusion of the additional data, we find that the storyline is clearer and more concise without the section on RNAP, while not diminishing its impact. Finally, upon discussion with our new co-authors, we have decided to use exact terminology and change PR to E_{raw} and S_{PR} to S_{raw} , respectively, as we believe these terms are clearer.

Specific concerns to be addressed:

More important:

1. The work sounds like it is a fundamentally new method although it is really adding one more photon

trajectory analysis in the existing framework of H²MM by Pirchi et al. <https://pubs.acs.org/doi/abs/> Therefore, the novelty of the method is in question.

We would like to thank reviewer #1 for this comment. Yes, the mathematics of the algorithm remains fundamentally the same. Nevertheless, taking into account the outcomes of the analyses of the data, including the data we added at the revision phase, we now show even more clearly the importance of using mpH²MM over the previous single parameter H²MM, especially when it comes to the decoupling of the intertwined FRET and photophysical dynamics. We hope that reviewer #1 can appreciate this point after carefully reading the revised manuscript. In one sentence, we show in the revised text that not using multi-parameter H²MM can even lead to inaccuracies as well as misinterpretations of the data. Therefore, we believe that the extension of the H²MM framework to multiple parameters is novel and quite important for practitioners who would like to quantitate FRET dynamics from smFRET measurements in a proper and accurate manner. All of that refers to the methodological and interpretational novelty.

Additionally, we add biological relevance as well. Using mpH²MM, on additional biomolecular systems that are more exploratory, we show that we can define the mechanism of their binding-coupled conformational dynamics quite accurately, which is a natural outcome of the improvements explained above.

Furthermore, one can claim that mpH²MM is applicable solely to nsALEX/PIE measurements. Although we made it clear in the original version that this is far from being the case, we did not provide examples on other types of multi-parameter smFRET-based measurements. We have now introduced section 2.4, where we show how to implement mpH²MM with μ sALEX setups. On the same line of thought, we have now also added more Jupyter notebooks that allow using mpH²MM not only on nsALEX and μ sALEX, but also on PIE-MFD measurements, taking into account not only E_{raw} & S_{raw} values but also $D_{\text{ex}}D_{\text{em}}$, $D_{\text{ex}}A_{\text{em}}$ & $A_{\text{ex}}A_{\text{em}}$ fluorescence anisotropies. These are all publically provided. We do not discuss any experimental PIE-MFD results in the main text, as it is out of the scope of the manuscript. However, we simply add this notebook for the benefit of the practitioners who would like to test using mpH²MM to analyze such results, and refer the reader to it from the discussion chapter.

Finally, we would also like to emphasize other novel contributions that come from this work: the introduction of the Integrated Complete Likelihood (ICL) the first extremum based statistical discriminator of state models. The original H²MM framework did not provide a statistical discriminator, and Dr. Lerner previously introduced the modified BIC (BIC'), which was based on threshold values.

2. In Sottini et al. <https://www.nature.com/> they also use a similar analysis to differentiate between subpopulations based on Viterbi paths where they also consider both ALEX and PIE setups. In this case, how does the current method presented in this manuscript provide more information or do so more robustly than the existing methods?

We would like to thank reviewer #1 for pointing this paper to us. From careful reading of this paper, it becomes clear that:

1. The authors of Sottini *et al.* used PIE of freely-diffusing molecules, when they wanted to assess slow FRET dynamics from burst recurrence analysis. However, the acceptor excitation pulsed laser in these experiments is not mentioned. From the statement on burst analysis, it is not clear how the A_{ex} Photon stream in this experiment was used. We believe the A_{ex} photon stream was used in these measurements, as it is typically used – as additional means of burst filtration. We therefore assume, reviewer #1 did not direct us for these freely-diffusing PIE smFRET measurements in Sottini *et al.*
2. The authors used their confocal-based setup to perform smFRET measurements on immobilized molecules. According to the methods' section, they used a single cw laser (532 nm), and hence it is

unclear to us, how was PIE or ALEX performed in this case, since the alternating acceptor excitation laser was not mentioned. Nevertheless, it is clear from the methods' section, that the single molecule fluorescence and FRET trajectories were analyzed using the Gopich & Szabo likelihood approach, which is why we believe reviewer #1 referred us to this part in Sottini *et al.* The likelihood approach taken to analyze single molecule FRET trajectories uses the smFRET traditional photon streams, $D_{ex}D_{em}$ & $D_{ex}A_{em}$, but there is no mention of using the additional photon stream outcoming of PIE/ALEX measurements, which is the $A_{ex}A_{em}$ photon stream. Without adding parameters that rely also on the $A_{ex}A_{em}$ photon stream, any likelihood approach would be similar to performing Pirchi *et al.* single parameter H²MM, which is not what we performed in this work – we performed multi-parameter based H²MM analyses, and show the advantage of using it over the single parameter version, which is, to the best of our knowledge not covered within Sottini *et al.*

3. We consider the possibility that perhaps the $A_{ex}A_{em}$ photon stream was used, but not reported in Sottini *et al.* In that case, we would be very happy to inspect it and compare it to the H²MM framework.
4. If indeed, it is the case as in the previous point (point 3): the key advantage of our method is that the stream of photons originating from acceptor excitation are directly incorporated into our MLE-based approach.

3. The mapping of E-S plots for PR and $S_{\{PR\}}$ relies purely on the Viterbi paths obtained based on the assumed number of subpopulations and the estimated parameters. Therefore, it is not clear how those error bars are obtained from Viterbi paths. Furthermore, error bars can only incorporate information about the assumed model. Therefore, it is not clear how the model uncertainty is incorporated in the estimates.

Following this comment from reviewer #1, we have added an additional supplementary figure to clarify how these plots are constructed, as well as additional color to figure 2 to highlight the state identification. As the text states, the model is selected based on the ICL. The bars displayed in the *Viterbi* E-S plots of figure 2, as stated in the figure legend, are actually the standard deviation of E-S values of all individual dwells based on *Viterbi* paths, since if standard error instead was used, the bars would be so small they would become visually invisible. We choose to show the standard deviation because it emphasized the dispersion of the E-S values of the dwells. Following reviewer #1's comment on uncertainty, we have also implemented additional methods to characterize the error, such as evaluating the log-likelihood of similar models, and error based on the variance of random sub samples, in a fashion similar to bootstrapping. Overall error estimates from the sub samples' procedure are smaller than the ones retrieved from *Viterbi* dwell-based analysis of standard deviations, and hence we would rather rely on the more conservative larger error values.

4. In previous work presented in Lerner *et al.*, “Transcription bubble of RNAP-..” , it was shown that the FRET dynamics are governed by 4 sub-populations. Here, in this work, they seem to recover the same subpopulations using the $S_{\{PR\}}$ values estimated from their Viterbi paths. Therefore, it is not clear what the new method fundamentally adds.

As stated, the S_{raw} values were derived directly, not through *Viterbi* paths. Here the use of Male and YopO serves to emphasize the need of direct incorporation of photon streams into the MLE-based approach. These can distinguish states with identical E_{raw} values that only differ in S_{raw} . With the addition of Male and YopO data, we decided to omit the RNAP data for the sake of clarity and brevity.

Less important:

5. The determination of the number of sub-populations relies in integrated complete likelihood (ICL) information. In other words, given the maximum likelihood estimates of the model parameters, the model is determined. However, this method does not propagate uncertainty on the possible models from the potential parameter estimation. Therefore, it is not clear how the error bounds are created for the estimates.

We understand the concerns reviewer #1 raised. However, the error bounds were not directly assessed; the quality of the fit is assessed by the ICL. The error is assessed through the closeness of the *Viterbi* paths to the predicted values of the model. We have now also provided a method to assess the error of a given model using the variance of random sub samples of the data. We believe these additions cover the subject of parameter value uncertainty estimations.

6. In all figures' panel (b), the y-axis label does not seem to be compiled successfully.

We would like to thank reviewer #1 for this comment. Now, this error has been fixed

7. In the end of “RNAP promoter open complex” section, the authors mention the use of fluorescent lifetimes to differentiate between the increase or decrease of the quantum yields of acceptor and donor dyes, respectively. Is it possible for the authors to explain how they extract these lifetimes from the experiments for non-experts in the field, at least a sentence stating that it is attributed to the pulse excitation setup?

As the RNAP data has now been removed from the manuscript, the mention of fluorescence lifetimes is no longer present in the main text. Nevertheless, we maintain in the supplementary data an explanation of how to extract fluorescent lifetime data, for the sake of practitioners who would be interested in extracting state-relevant fluorescence decays. In addition, we provide in the SI a detailed description of how to use parameters that are not centrally distributed, such as the photon nanotimes in time-resolved smFRET, and how to transform them into centrally distributed parameters, to be used as additional parameters within the mpH²MM framework.

8. The authors directly start the analysis of experimental data however, I suggest that they showcase their method on simulated data where the plain H²MM fails. This would help in easily seeing the difference of their method with the existing competing methods.

We thank reviewer #1 for this excellent suggestion. Now this has been added to the manuscript. We refer reviewer #1 to the added first subsection in the Results chapter, where we have implemented such simulations, and clearly demonstrate how single-parameter H²MM fails in cases where multi-parameter H²MM is fully capable of extracting all relevant parameters and states.

9. On page 2, the authors already introduced PR for proximity ratio and on page 3, they pre-fer to go back to “proximity ratio”. I suggest them to stick to their convention consistently.

After further discussion, we have changed PR to E_{raw} , similarly, S_{PR} to S_{raw} , therefore this confusion should no longer occur. Similarly, we no longer refer to proximity ratio, but rather to raw FRET efficiency, following the convention from Lee *et al.* 2005.

10. Can you provide inline equation for S_{PR} ?

This is provided in supplementary equation S2 (now renamed S_{raw} for terminology accuracy). We have also added a reference to this equation in the main text when S_{raw} is first introduced, in the introduction chapter.

Reviewer #2 (Remarks to the Author):

Summary

Harris et al. introduce multi-parameter H2MM (mpH2MM) and provide ‘proof of concept’ on smFRET data using a DNA hairpin and the transcription bubble formed with RNA Polymerase (RPO). They propose the use of the integrated complete likelihood (ICL) to determine the most likely state-model. Using the new mpH2MM they are able to additionally discriminate dark acceptor and donor sub-populations in a DNA hairpin. For the transcription bubble data, analysis using mpH2MM additionally enables the recovery of stoichiometry information. While the paper presents major advances in the speed and functionality of photon-by-photon H2MM analyses for smFRET, there are a few issues to address before publication in Nature Communications can be recommended.

We would like to thank reviewer #2 for his objective and overall positive assessment of our work.

Dear reviewer #2.

Following your in-depth analysis of our work, we have added developments as well as experimental details, to fully answer the concerns you and reviewer #1 raised. These include:

1. Added novelty by
 - a. Showing the usefulness of the multi-parameter approach to properly and accurately decouple photophysical dynamics from FRET dynamics that are typically intertwined in smFRET experiments. Comparisons of the results against the single parameter H²MM prove this point, now on simulations, experiments on well-controlled biomolecular systems and also on more exploratory systems.
 - b. showing the biological relevance of using mpH²MM to distinguish induced fit and intrinsic dynamics in the new experiments with MalE and YopO, respectively
2. Provide a work-around for μ sALEX setups, and demonstrate its use with YopO
3. Significantly updated the notebooks and text to make them more understandable to non-experts

We should also note that with the addition of the MalE and YopO data, we chose to omit the RNAP part for clarity and brevity. Finally, upon discussion with our new co-authors, we have decided to change PR to E_{raw} and S_{PR} to S_{raw} for the sake of terminology accuracy.

Below you can find a point-by-point response to all the concerns you have raised. Again, thank you for the thorough analysis.

Detailed overview

Multi-parameter H2MM as presented here allows access to a further dimension for analysis of biomolecule dynamics using smFRET. The authors’ description of their implementation of the H2MM code in C as ‘dramatically faster’ undersells the improvement – the implementation completely changes the ability to perform H2MM analysis on large/complex datasets.

We would like to thank reviewer #2 for this view. Indeed, we think so as well and hope that the revised text will highlight this point.

While this improvement is game-changing, the utility of mpH2MM over ‘spH2MM’ is not well demonstrated. While the discrimination of dark acceptor/donor sub-populations for the DNA hairpin perhaps sheds some light on ‘photo-physical transitions’ this is not relevant from a biomolecular perspective and was not used to provide any new information about the system’s dynamics. Similarly, for the RPO, while the authors present an avenue to elucidate the underlying photo-physical mechanism

causing a decreased SPR for one of the sub-populations, the proposed use of this additional information to model the structure is not demonstrated to be necessary or effective.

We are thankful for this comment from reviewer #2. We recognize that our previous examples did not properly emphasize the relevance of distinguishing photophysical dynamics from conformational dynamics. Our new data, however, emphasizes how it is necessary. Specifically 1) our simulations show how incorrect state and transition rate assignments can occur without incorporation of the acceptor excitation photon stream, and 2) this same effect is seen clearly with the spH²MM vs. mpH²MM state assignment in the MalE data. We believe that the proteins can affect S_{raw} but as the RNAP data is now replaced with MalE and YopO, which do not show evidence for this, such discussion has been removed from the main text, although discussion of fluorescence lifetimes remains in the supplementary material.

Major revisions

1) mpH²MM as implemented here is extremely powerful and likely to supplant current smFRET dynamics analysis methods. While the use of multi-parameter analysis has been shown to elucidate some further photo-physical information its usefulness for ‘conformational transitions’ was not well supported. The use of two previously well-characterized systems limits the potential impact of this work. In the conclusions it is stated that mpH²MM increases the ‘information content of the results and the sensitivity of the H²MM algorithm’ – a discussion about the latter comparing the results derived from spH²MM and mpH²MM is missing. Providing this is strongly recommended.

We understand the important point reviewer #2 made. Following that, we have added the MalE and YopO data in which we actually expose the mechanism by which conformational dynamics is coupled to binding in two new protein systems: distinguish induced-fit conformations from intrinsic dynamics. In addition, with YopO, we report an accurate description of intrinsic dynamics in YopO undergoing conformational dynamics at the tens of microseconds without needing the assistance of lifetime-based data.

Both examples show how important using mpH²MM could be for assessing conformational dynamics from smFRET measurements.

We hope this satisfies the novelty side of the application of mpH²MM, requested by reviewer #2.

For example, a key question the authors need to address in the discussion is:

For DNA hairpin if BVA shows there is no within-burst FRET dynamics what is the point of the ‘dark acceptor’ state? Couldn’t this be filtered out using a dual colour burst search (DCBS)? If a DCBS is used (or a minimum AexAem threshold) is mpH²MM any better than spH²MM at determining the ‘biologically relevant’ states?

We would like to thank reviewer #2 for this interesting point. Firstly, we would like to point out to reviewer #2 that the text and the figure do show a signature of within-burst dynamics. The only question that is left is whether this is within-burst FRET dynamics, between the two FRET-active sub-population, or perhaps acceptor-blinking dynamics. Regardless, the BVA plot does report within-burst dynamics.

In the case of a system that does not exhibit a signature of within-burst FRET dynamics, it is possible that searches/filters such as DCBS will remove information such as the dark acceptor instances.

However, our new Male analysis uses DCBS, and in the supplementary figures we also present DCBS for all of the DNA hairpin data (not only at $[\text{NaCl}] = 300 \text{ mM}$). What we find is that DCBS is not always sufficient to remove blinking transitions, only the ones that are slower than burst durations. Thus, even in such data, dark acceptor states still occur, but within bursts, hence the potential to observe a BVA signature of within-burst dynamics that is not necessarily related to FRET dynamics. If BVA were not to show within burst dynamics, then H²MM single- or multi-parameter is not generally recommended. H²MM is primarily for analysis of data that does show within-burst dynamics.

2) While we agree in principle with the authors' own suggestion that the 'most obvious application' of mpH²MM being in multi-color smFRET measurements, this is not demonstrated. The considerable commitments required to perform these experiments is appreciated and we would not require these for publication, however, a demonstration of mpH²MM's utility using simulated data would be greatly add to the impact of the manuscript.

We would like to thank reviewer #2 for this comment. Now we include a Jupyter notebook that provides application of mpH²MM to FRET & anisotropy measurements, (better known as part of the PIE multi-parameter fluorescence detection, PIE-MFD), and point the reader to this in the discussion chapter. We hope that this satisfies the request of reviewer #2 for demonstrating the general applicability of mpH²MM, even if it is not for 3-color FRET. We explain in the discussion chapter, that this application can be adapted easily to multi-color smFRET, by the same treatment of the available photon streams – while in PIE-MFD there are 6 streams for 5 ratiometric parameters (i.e., E_{raw} , ratiometric S_{raw} and for $D_{\text{ex}}, D_{\text{em}}, D_{\text{ex}}, A_{\text{em}}$ and $A_{\text{ex}}, A_{\text{em}}$ fluorescence anisotropies), in 3-color smFRET, there are 3 streams for 3 ratiometric E_{raw} values. The change between the two is the difference between using polarizing beamsplitters and adding dichroics on the same multiple point detectors.

3) For this work to be impactful it needs to be accessible and usable by non-computer scientists and consequently we make the following comments: the Jupyter notebooks as uploaded have runtime errors from the authors – they should show complete successful outputs from running in their hands. The 'Multi-parameter photon-by-photon hidden Markov modeling dataset' (<https://doi.org/10.5281/>) is uploaded as a series of files - we strongly recommend re-uploading to maintain the directory as the notebooks currently reference a non-existent hierarchy. Also, it is not possible to proceed through the notebooks as some steps reference inputs generated with different names to those output in previous steps thereby raising errors and the code cannot run without modification. Where a name change is required to match outputs this should be commented in the code. The notebooks in the second repository 'harrisd/H2MMpythonlib: H2MM_C' (<https://doi.org/10.5281/>) do not use the C implementation of the H2MM code and cannot run as the non-C H2MM module (py script) is replaced by the C version. After some modifications to the notebooks we were able to run the H2MM_C and as mentioned it represents a dramatic improvement in the speed of analysis. However, we have not been able to proceed beyond the 'viterbi_sort' method which repeatedly kills the kernel. Ideally everything should be uploaded in a way that others can run, including a list of minimum hardware requirements (if any) as well as details on the python environment and dependencies - as we were unable to perform the full analysis in our hands.

Following the important comment of reviewer #2, we have thoroughly updated the Jupyter notebooks, and made numerous additions (color-coded labels above cells, for instance making cells where burst search etc. parameters are set, hyperlinks to various tutorials on Python and to specific parts of the Jupyter notebook).

The zenodo repository has been updated, and now for HP3 measurements, it will work as requested, so long as the HDF5 files are in the same directory as the Jupyter notebook. For Male and YopO there

were many HDF5 files, we choose to package them into zip containers and have the notebook reference subdirectories.

The issue with *viterbi_sort* killing the kernel was a bug in the code that is OS and system dependent related to the maximum number of threads the computer can handle at a time. We have fixed this issue, and the code in the updated repository should run without issue (follow the updated zenodo link and install the new version of H2MM_C).

We also include in the supplementary information the details of the laptop we used, and what other systems we have used to test the code, and we include the time it took each notebook to run, and how long it took to fit each dataset within the notebooks. We hope this is sufficient for benchmarking and minimum system requirements.

Minor revisions:

The authors present the BIC and ICL as discriminators for the number of states. Can they please comment on how much of difference in the ICL is needed to confirm adding the extra state is significantly better? Looking at Figures 2C and 3C this is not immediately obvious.

We generally use ICL purely as an extremum parameter, i.e., we do not have a clear minimum difference between ICL to consider a model significantly better. Upon further testing, we find the most reliable method is to compare both ICL and BIC' results. The text in the new simulations subsection is particularly relevant to your point:

“Yet, there are instances in the simulated data, and in real data sets we describe later, where the selection of the most likely state model based on ICL is of a model with too few states, relative to our prior knowledge of the system. Therefore, we always consider the ICL first, then BIC', and take into account the prior knowledge of the system when selecting the most likely state model (see supplementary section S2 for expanded discussion, supplementary figure S2).”

The authors need to be clear as to exactly how the plots in Fig 2e,f and 3e,f are generated, if any burst search is applied etc. This additional detail could appear in the figure legend or the Materials and Methods.

The following explanations have been added to the figure legend *“Consecutive photons with the same state are considered as a single dwell, E_{raw} and S_{raw} values are then calculated as in equations S8 and S9, respectively.”* Additionally, in the main text, we point the reader to the new supplementary figure S6 in the Viterbi portion of the HP3 results, and at the end of the figure legend, which graphically shows how these plots are created.

Why will the new method not work for microsecond ALEX? How much extra effort would it be to make this work? Such an expansion would be extremely impactful increasing the reach of the new analysis to a much wider userbase who don't use pulsed lasers.

We thank reviewer #2 for encouraging us to pursue μ sALEX experiments. This has resulted in us including the new YopO data (Section 2.4), which is μ sALEX, as described in the new text, a time shift must be applied to $A_{ex}A_{em}$ photons to make them overlap in time with D_{ex} photons. We do note, both here and in the new text that one must be careful in interpretation of this data as it comes to S_{raw} values (not influencing the accuracy of the E_{raw} values), to avoid artefacts due to the alternation period. As we note in the new text, without application of this shift, the alternation period is detected.

‘occurring at times from picoseconds to seconds and slower[3].’

- Should state rates rather than times if saying ‘slower’ or changed to longer changed

‘This is indeed the case in the coupling of catalysis with substrate binding domain dynamics in some enzymes[4, 5], the dynamics of the DNA bubble in transcription initiation to support transcription start site selection[6, 7], involved in DNA mismatch repair[8], to facilitate protein translocation across membranes[9], assisting chaperone action[10], involved in the allosteric regulation of the AAA+ disaggregase[11], influences the selectivity in ABC importers[12], and many other important bio-processes, in which structural dynamics is coupled to biological function or influences it[1,2].’

- This single sentence needs re-writing for clarity probably into several shorter sentences.

This sentence has been rewritten and divided, it now reads:

“Examples include coupling of catalytic activity to domain dynamics in some enzymes[4,5], the dynamics of the DNA bubble in transcription initiation to support transcription start site selection[6,7], DNA mismatch repair[8], protein translocation[9], chaperone action[10], the allosteric regulation of the AAA+ disaggregase[11], active membrane transport[12-17], and many other important biochemical processes, in which structural dynamics is coupled to or influences biological function[1,2].”

‘Nevertheless, while advanced smFRET setups often detect multiple parameters beyond the simple proximity ratio, such as in alternating-laser excitation (ALEX; also known as pulsed-interleaved excitation, PIE)[28, 29] or in multi-color smFRET-based measurements[30–37], H2MM in its current iteration only reports on the proximity ratio of a single donor-acceptor pair of dyes. Here, we introduce multi-parameter H2MM (mpH2MM), which enables incorporation of multiple parameters, through additional photon streams in nanosecond ALEX (nsALEX)(a.k.a. PIE)[28, 29] experiments.’

- Referencing both ALEX and nsALEX as ‘a.k.a. PIE’ is confusing and can be misleading to broad readership of the journal. Be clear about the nomenclature here - nsALEX is indeed PIE, but ALEX can be microsecond ALEX (using continuous wave lasers) so is not necessarily PIE (which always uses shorter pulses).

We have revised how we introduce these methods. First, we introduce ALEX

“Nevertheless, while advanced smFRET setups often detect multiple fluorescence parameters beyond the simple intensities, such as in alternating-laser excitation (ALEX)[60, 61] ...”

and then introduce the two “flavors” and give the alternate name of nsALEX (PIE) a few sentences later.

“We demonstrate this concept with two types of ALEX experiments: microsecond ALEX (μ sALEX) and nanosecond ALEX (nsALEX; known also as pulsed interleaved excitation, PIE)[60, 61]”

‘However, after using mpH2MM, it turns out that the four-state model is the state-model that best explains all of the data, including the acceptor-excitation data’

- ‘it turns out that’ reads too informally – reword or can remove transition altogether

The entire paragraph has been rephrased to be more clear. It now begins with:

“Analyses of this data with spH²MM and mpH²MM show different patterns in the ICL values of the state-models. The ICL is minimized for spH²MM models for a two-state model, while it is minimized for a four-state model when using mpH²MM.”

‘Using two-color microsecond ALEX experiments, Robb et al. has shown’

- Suggest rewording to ‘have shown’ per conventional usage

As explained before, with the additions of MalE and YopO data, this section was removed for brevity and clarity.

‘Multi-parameter H2MM increases both the information content of the results, and the sensitivity of the H2MM algorithm to differences in the photon streams’

- photon streams
- changed

As ALEX or PIE setups are now ubiquitous

- Not sure about the term ubiquitous here.

Changed to “*commonly used*”

‘Conversion of the emission probability matrix into useful $PR_{i,j}$ values’

- No previous mention or declaration of what $PR_{i,j}$ values are

We have rephrased this to:

“If qualitative tests indicate that such a system is undergoing within-burst dynamics, mpH^2MM is well-suited to extract the transfer efficiencies relevant to the underlying dynamically-interconverting sub-populations.”

‘Experimenters using other platforms must utilize their knowledge of the peculiarities of their data’

- Suggest changing ‘peculiarities’ to particulars

changed to “*fine details*”

‘We use the H2MM algorithm (both single- and multi-parameter) to test how well 2, 3 or 4 state-models describe the photon data’

- 2 is subscript rather than superscript

changed

This also allows bursts to be classified by which and how many states are present.

- bursts to be classified

changed

Various spelling/typographic errors in the notebooks.

We have edited and substantially updated the notebooks

Reviewers' Comments:

Reviewer #1:

Remarks to the Author:

The authors modified their originally submitted paper and carefully addressed my questions and comments. Overall, I am satisfied with the answers that they gave and I don't have further major questions.

Remaining minor concerns:

1 - In the introduction, on Page 3, they mentioned that single parameter H2MM treats the single photon data however, this is not correct, there are still bins that are limited by the smallest interphoton arrival time as it is noted on the paragraph prior to Eq 4 in that Pirchi et. al paper. I suggest the authors correct their language on that aspect.

2- Can the authors provide comments in their discussions on whether TIRF community might benefit from their method?

Reviewer Response

Authors responses are shown in blue and quotations of new content are highlighted in yellow

Reviewer #1 (Remarks to the Author):

The authors modified their originally submitted paper and carefully addressed my questions and comments. Overall, I am satisfied with the answers that they gave and I don't have further major questions.

We thank the reviewer for his kind positive response to our revision. We have added sections addressing all his remaining concerns. We hope these will be satisfactory, and publication of our work can proceed.

Remaining minor concerns:

1 - In the introduction, on Page 3, they mentioned that single parameter H2MM treats the single photon data however, this is not correct, there are still bins that are limited by the smallest interphoton arrival time as it is noted on the paragraph prior to Eq 4 in that Pirchi et. al paper. I suggest the authors correct their language on that aspect.

We have added the following qualification to the text:

other than the clock time of the acquisition card (e.g.) 50 ns for nsALEX, 12.5 ns for μ sALEX

While Pirchi *et al.* state that they choose a bin size, in our implementation we use the resolution of the data acquisition equipment and software itself as that "bin" and thus our claim that we do not lose any information as a result of binning remains true.

2- Can the authors provide comments in their discussions on whether TIRF community might benefit from their method?

We have added the following paragraph to the discussion:

The success of integrating the acceptor excitation stream into our analysis, suggests that a similar approach could also be employed in camera-based smFRET applications. Alternating laser excitation, and HMM algorithms are commonly employed in analysis of data, although the information of the acceptor excitation stream is discarded in these analyses, and only used for truncating trajectories upon acceptor bleaching. The present mpH²MM algorithm is inappropriate for such data as the information is based on intensity and a constant frame rate, instead of single photon arrivals with variable interphoton times. Alternatively, the introduction of SPAD arrays should allow analysis of immobilized molecules with single photon precision, which would allow for analysis of such data directly using mpH²MM.

We hope this discussion is sufficient.

Editorial Response:

We have made all additions requested by the editors. Of note we have added n values for the number of dwells used in calculating the standard deviations shown in all figures.

We have also added the code availability section and details on versions of software with links to their respective repositories.

We have also adjusted the naming of “supplementary section” to “supplementary note” in both main text and supplementary information and removed the “S” prefix from numbers.

Finally, we added the following text to the end of the introduction, explaining what we mean by multi-parameter:

It should also be noted that we use the term parameter in multiparameter H2MM to refer to parameters derived from ratios of sums of photons in different photon streams (e.g. E and S). These are distinct from state model parameters (e.g. rate constants, mean E).